# The NF-κB Factor Relish maintains blood progenitor homeostasis in the developing *Drosophila* lymph gland

**Parvathy Ramesh**[1⊙], **Satish Kumar Tiwari**[1⊙], **Md Kaizer**[1⊙], **Deepak Jangra**[1], **Kaustuv Ghosh**[1], **Sudip Mandal**[2], **Lolitika Mandal**[1]*

**1** Developmental Genetics Laboratory, Indian Institute of Science Education and Research Mohali (IISER Mohali), Punjab, INDIA, **2** Molecular Cell and Developmental Biology Laboratory, Department of Biological Sciences, Indian Institute of Science Education and Research Mohali (IISER Mohali), Punjab, INDIA

⊙ These authors contributed equally to this work.

* lolitika@iisermohali.ac.in

**Data Availability Statement:** All relevant data are within the manuscript and its Supporting Information files.

## Abstract

Post-larval hematopoiesis in *Drosophila* largely depends upon the stockpile of progenitors present in the blood-forming organ/lymph gland of the larvae. During larval stages, the lymph gland progenitors gradually accumulate reactive oxygen species (ROS), which is essential to prime them for differentiation. Studies have shown that ROS triggers the activation of JNK (c-Jun Kinase), which upregulates fatty acid oxidation (FAO) to facilitate progenitor differentiation. Intriguingly, despite having ROS, the entire progenitor pool does not differentiate simultaneously in the late larval stages. Using expression analyses, genetic manipulation and pharmacological approaches, we found that the *Drosophila* NF-κB transcription factor Relish (Rel) shields the progenitor pool from the metabolic pathway that inducts them into the differentiation program by curtailing the activation of JNK. Although ROS serves as the metabolic signal for progenitor differentiation, the input from ROS is monitored by the developmental signal TAK1, which is regulated by Relish. This developmental circuit ensures that the stockpile of ROS-primed progenitors is not exhausted entirely. Our study sheds light on how, during development, integrating NF-κB-like factors with metabolic pathways seem crucial to regulating cell fate transition during development.

## Author summary

The determination of cell fate in stem/progenitor cells is crucial for normal development and various pathophysiological conditions. Our research focuses on the *Drosophila* larval blood-forming organ, the lymph gland, as a model to gain further insights into this intricate process. Interestingly, while the entire progenitor pool within this organ is enriched in Reactive Oxygen Species (ROS), which is a known trigger for differentiation, they do not differentiate entirely. We show that Relish, a component of the Immune Deficiency Pathway (IMD), plays a role in preventing premature differentiation of the progenitors. Relish hinders TAK1 (transforming growth factor-β-activated kinase 1), thereby reducing

**Funding:** This study was supported by DST-SERB Grant CRG/2020/000511 to SM and DBT, Wellcome-Trust India Alliance Senior Fellowship [IA/S/17/1/503100] and DST-SERB Power Grant SPG/2021/000122 to LM. Thanks to IISERMohali Institutional support to PR, SM and LM, Council of Scientific and Industrial Research (CSIR, INDIA) fellowship to SKT and MK, PMRF to DJ, KVPY fellowship to KG. The funders had no role in study design, data collection and analysis, decision to publish, or preparation of the manuscript.

**Competing interests:** The authors have declared that no competing interests exist.

JNK activation downstream of ROS. This inhibition on JNK delays progenitor differentiation. Our study reveals an interplay between developmental signaling and metabolic factors that govern the fate specification and maintenance of blood progenitors. Given that the process of blood development in flies shares several similarities with mammalian hematopoiesis, including the presence of high ROS in the myeloid progenitors, it would be worth exploring whether similar interactions are at play in vertebrate hematopoiesis.

In addition, it would be fascinating to investigate whether similar coordination between metabolic and developmental signals regulates the differentiation of stem/progenitor cells in other biological systems.

## Introduction

The site for definitive hematopoiesis in *Drosophila* is the lymph gland, a multi-lobed, larval blood-forming organ [1–5]. During larval development, the prohemocytes within this organ proliferate and differentiate. With the onset of pupation, the entire cohort of hemocytes is released into circulation to take care of post-larval requirements [6,7].

The mature lymph gland has a well-characterized anterior lobe/primary lobe with three distinct zones and a niche (Posterior Signaling Center) crucial for maintaining the bulk of the progenitors [5,8]. The medially located heterogeneous progenitor cells [9–11] constitute the medullary zone (MZ) at the core and distally located rim of intermediate progenitors (IZ), while the differentiated hemocytes in the periphery of the organ define the cortical zone (CZ) of the primary lobe (**Fig 1A**) [3].

Although the lymph gland progenitors proliferate to increase their number right from the early instar, robust differentiation is noticed only in the third instar stage [3]. It is intriguing how this reserve population within the lymph gland is intercepted from precociously inducting into the differentiation program. Studies have shown that developmentally regulated, moderately high Reactive Oxygen Species (ROS) levels in the blood progenitor cells of the *Drosophila* mature lymph gland elicit c-Jun Kinase (JNK) to facilitate their differentiation [12]. Recent studies have also shown that the JNK-FAO axis is essential for progenitor differentiation [13].

We wondered that if ROS triggers differentiation, then why doesn't the pool of progenitors differentiate entirely? Why does the differentiation of the progenitors initiate at the rim of MZ despite the entire MZ being enriched in ROS? What shields the progenitors from entering the differentiation program even though they are enriched in ROS? Our interrogation identified nuclear factor-κB (NF-κB) transcription factor Relish, a component of the immune deficiency (Imd) pathway, as a key molecule that prevents the progenitor pool from differentiating precociously.

Although the Imd/IMD pathway has been studied intensively in immunity and inflammation, its role during fly development has only recently been unraveled. Studies have elucidated the non-inflammatory roles of Relish (Rel) in *Drosophila* neurodegeneration [14–16], apoptosis [17] and autophagy in salivary gland cells [18]. Interestingly, in the absence of infection, persistent activation of IMD in *Drosophila* intestinal stem cell progenitors increases the frequency of division in Intestinal Stem Cells [19]. Intrinsic Nf-κB activity is shown to be essential in stem cells for repairing damaged gut epithelia in *Drosophila* adult [20].

During development, Rel in the hematopoietic niche of the lymph gland is crucial for its functionality. Absence of Relish from the niche induces JNK-dependent cytoskeletal rearrangement. The aberrant cytoskeletal rearrangement perturbs the proper delivery of Hedgehog from the niche to the adjoining progenitors, leading to their precocious differentiation.

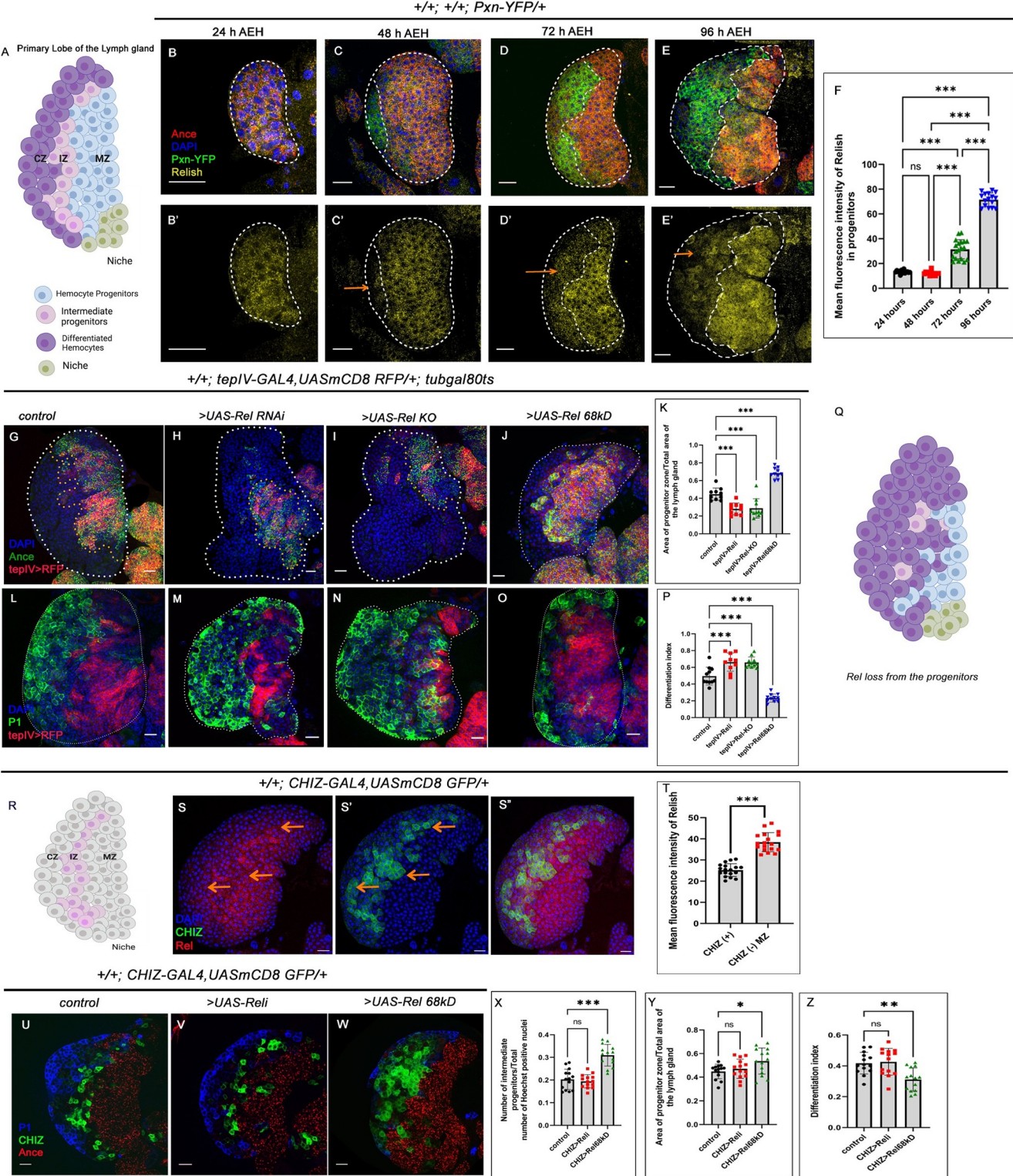

**Fig 1. Loss of Relish from the progenitor cells causes precocious differentiation.** (A) Scheme of the Primary lobe of the third instar larval lymph gland with different cell types present in it. (B- E') Relish (yellow) is enriched in progenitors (Ance: red), whereas downregulated in the differentiated cells (arrows, Pxn: green). (F) Mean fluorescence intensity of Rel expression in the progenitors Fig 1B-1E' (lymph glands n≥12, P-values from left to right <0.923, <0.001, <0.001, <0.001, <0.001 and <0.001 respectively). (G-J) Compared to control (G), Rel loss of function by RNAi (H) or progenitor-specific knockout of Rel by CRISPR/Cas9 system significantly reduces the number of progenitors (Ance, I). Rel overexpression expands the progenitor zone (Ance, J). (K) The ratio

of the area of progenitors to that of the total area of the primary lobe for the genotypes G-J (lymph glands n = 10, P-values from left to right <0.001, <0.001 and <0.001 respectively). (L-O) Rel loss from the progenitor leads to increased differentiation (P1: green, M-N) compared to that of the control (L), whereas Rel overexpression affects progenitor differentiation (P1: green, O). (P) Differentiation index for genotypes L-O (lymph glands n≥10 for each genotype, P-values from left to right <0.001, <0.001 and <0.001 respectively). (Q) Schematic representation of the phenotype obtained upon rel loss from the progenitors. (R) Scheme represents the IZ (pink) within the mature lymph gland. (S-S") Co-labeling of Rel with CHIZ>GFP reveals that the expression of Rel is down regulated in CHIZ expressing cells. (T) Quantitation of the results from S-S". (lymph glands n = 18 for mentioned genotype, P-value < .001). (U-W) Downregulation of Rel from the intermediate progenitor did not affect their number (CHIZ: green, V), nor differentiation (blue, V) or progenitor (red, V) area compared to that of the control (U), whereas overexpression of Rel causes an increase in the number of intermediate progenitors (W). (X) The ratio of the intermediate progenitors to the total number of cells in the primary lobe for the genotypes U-W. (lymph glands n = 14 for each genotype; P values: 0.843 and <0.001 respectively). (Y) The ratio of the area of progenitors to that of the total area of the primary lobe for the genotypes Fig U-W (lymph gland n = 14, P-values from left to right = 0 .768 and = 0 .027, respectively). (Z) Differentiation index for genotypes Fig U-W (lymph gland n = 14, P-values from left to right = 0.947 and <0.003 respectively). Genotypes are as mentioned. Each dot in the graph represents individual values, and Data is expressed as mean ± SD. Statistical analysis: Tukey's multiple comparisons tests. Unpaired t-test with Welch's correction for T. P-Value of <0.05, <0.01 and<0.001, mentioned as *, **, *** respectively. Scale bar: 20μ, DAPI: nucleus.

Strikingly, Relish is downregulated in the niche during bacterial infection to facilitate an early dispersion of lymph gland resident hemocytes into circulation. The dynamic expression of Relish endorsed that the developmental pathway gets recalibrated in the hematopoietic niche to combat high bacterial infection [21].

Throughout development, the strong presence of Rel in the hemocyte progenitors caught our attention. Employing expression analyses, genetic and pharmacological approaches, we show that Rel inhibits activation of JNK and thereby shields off the core progenitors from differentiating. Overexpression of Rel resulted in a halt in the differentiation program.

The metabolic input essential for triggering the differentiation of the progenitors is ROS. ROS evokes JNK, which transcriptionally regulates Carnitine palmitoyl transferase I (CPTI) /*withered* (*whd*), the rate-limiting enzyme of FAO, to initiate the differentiation of the progenitors [13]. The outcome of the above study demonstrated how cellular signaling machinery collaborates with the metabolic cue to enable the differentiation program.

However, it does not take into account how the stockpile of ROS-enriched progenitors is maintained. Here, we show that the metabolic input in the form of ROS is monitored by a developmental signal, which shields the bulk of ROS enriched progenitors from inducting into the differentiation process. As a result, the differentiation program is restricted to the progenitors housed in the outer rim of the Medullary Zone.

Our study elucidates how the integration of immune-developmental and metabolic axis is essential for regulating myeloid-like progenitor homeostasis during development.

## Results

### Loss of Rel from the hemocyte progenitors causes precocious differentiation

The primary lobe of third instar lymph gland consists of the heterogeneous progenitor, which can be visualized by different markers. Angiotensin (Ance), Domeless (Dome) and Thioester-containing-protein 4 (TepIV) are expressed by the progenitors [1,11]. The intermediate progenitors are detectable as Dome and Peroxidasin (Pxn+) [11] or Dome and Hemolectin (Hml+) and by CHIZ-Gal4.UAS GFP expression [22]. As evident from the figure, MZ progenitors (Ance+Pxn-) express high levels of Relish throughout larval development compared to the CZ (**Fig 1B–1E' and 1F**). The early onset of Rel expression, along with the differential labeling in the mature lymph gland, prompted us to investigate the role of Rel in blood progenitors during development.

A progenitor-specific downregulation of Rel (*tepIV >UAS-Rel* RNAi) revealed a significant reduction in the progenitor pool (Ance) (**Fig 1G–1H** and **1K**). This observation was further

strengthened when CRISPR-mediated knockout of the Rel function from MZ also led to a decline in progenitor number (**Fig 1I** and **1K**). The loss in progenitor number was accompanied by an increase in the differentiated cell population (plasmatocyte: P1 Nimrod [23], **Fig 1L–1N, 1P** and **1Q**) and Crystal cell: Hindsight, Hnt (**S1A, S1B** and **S1D Fig**). Despite precocious differentiation observed upon loss of Relish from the progenitors, the lack of lamellocytes (a specialized class of hemocytes which is generated upon immune challenge, **S1E–S1F Fig**) implicated that Rel loss does not evoke an immune response.

Interestingly, the lymph gland analysed from germ-free/axenic larvae (devoid of commensal microflora, **S1G Fig**), showed no significant change in the differentiation index when compared to control (**S1H–S1J Fig**). Moreover, the expression of Relish in the progenitors also remained unaltered in an axenic condition (**S1K–S1K" Fig**). Based on these results, we can infer that the commensal microflora does not elicit Relish expression and activation in the progenitors, suggesting a developmental role of NF-κB in blood progenitors. Moreover, in sync with previous report [21], upon bacterial infection there was a decrease in Relish expression from the hematopoietic niche compared to uninfected larvae (**S1L–S1M' Fig**). In contrast, Relish expression was not affected in the lymph gland progenitors of infected larvae compared to the sham and control lymph glands (Compare **S1N–S1N', S1O–S1O', S1P-S1P' and S1Q Fig**). This set of results implies that in contrast to the niche-specific Rel expression, the progenitor specific expression of Relish is not susceptible to the pathophysiological state of the organism but instead seems to serve as a developmental cue.

A similar decline in progenitor number (Ance+, **S1R–S1R' and S1R'" Fig**) and increased differentiation (P1+, **S1S–S1S' and S1S'" Fig**) were evidenced upon downregulating Rel function by another validated progenitor-specific Gal4: domeless [24–27]. In contrast, overexpression of Rel (by *UAS-Rel 68* kD: which represents the 'active' cleaved product of full-length Rel) with either of the drivers leads to a halt in the differentiation of the progenitor pool (Ance+, tepIV+: **Fig 1G, 1J and 1K**, and Ance+, Dome+: **S1R, S1R" and S1R'" Fig**) and (P1: **Fig 1L, 1O and 1P**, Dome+, P1, **S1S, S1S" and S1S'" Fig**). Interestingly, the number of intermediate progenitors (Dome>RFP, Pxn-YFP) [1, 13, 22, 28] declines upon loss of Rel (arrows, **S1T-S1T' and S1T'" Fig**) while the sustained expression of Rel leads to their abundance (arrows, **S1T, S1T" and S1T'" Fig**).

To endorse the above observation, we first carried out Rel expression analysis with a validated marker of IZ: CHIZ-GFP (**Fig 1R**). Interestingly, the Rel expression is downregulated in the CHIZ- expressing cells compared to the enrichment evidenced in MZ cells (**Fig 1S–1T**).

To explore the role of Rel, if any, in the IZ, we downregulated Rel expression specifically in the intermediate progenitor (CHIZ-GAL4>UAS-GFP), which did not affect IZ (**Fig 1U–1V** and **1X**), neither MZ (**Fig 1Y**) or CZ (**Fig 1Z**). On the other hand, ectopic expression of Rel in the CHIZ- expressing cells led to a robust increment in IZ number (Compare **Fig 1U** with **1W** and **1X**), a moderate increase in MZ (**Fig 1Y**) and a decline in the number of differentiated cells (**Fig 1Z**). Therefore, the low level of Rel in the IZ evidenced during development is crucial for facilitating the differentiation of IZ cells.

This set of experiments collectively illustrates that Rel is necessary and sufficient to maintain the pool of hemocyte progenitors.

In the late third instar stages, the primary lobe progenitors in the lymph gland experience a G2/M arrest [11] before the onset of differentiation. We employed Fly-FUCCI [29] to explore whether the defects in differentiation observed upon Rel perturbations result from an altered cell cycle of the progenitors. Fly-FUCCI relies on fluorochrome-tagged degrons from the Cyclin B and E2F1 proteins, which are degraded by the ubiquitin E3-ligases APC/C and CRL4/Cdt2 during mid-mitosis or the onset of the S phase, respectively. Cell cycle analysis revealed that loss of Rel function from the progenitors disrupts G2/M arrest, and the residual

progenitors are pushed towards the S phase before their differentiation (**S2A–S2B** and **S2D–S2D' Fig**). The relief from G2-M arrest and re-entering into S phase is an event that precedes differentiation during normal development [22]. The rate of this event escalates upon loss of Rel from the progenitors.

Intriguingly, in Rel overexpression, compared to control where cells are in G2-M arrested condition, a significant number of cells were in the S phase (Compare **S2A** and **S2C** and **S2D" Fig**). Studies have shown that the progenitors proliferate before achieving a G2-M halt [11,13,30]. The abundance of cells in the S phase indicates the delay in slowing the ongoing proliferation upon Rel overexpression. The thymidine analog EdU incorporation assay further endorsed the increase in the proliferative phase. Compared to the control, where there was a negligible number of EdU positive cells in the *tepIV* expressing progenitors, loss of Rel and Rel overexpression resulted in increased incorporation of EdU, signifying the abundance of the S phase in the progenitors (**S2E–S2H Fig**).

Together, this set of experiments suggests that perturbation in Rel expression affects the cell cycle status of lymph gland progenitors.

## Rel loss from the progenitors leads to activation of JNK

Hemocyte progenitors of the primary lobe of the lymph gland generate high levels of physiological Reactive Oxygen Species (ROS), which serves as the primary signal for their differentiation [12]. Co-labeling of Rel with gstD1-GFP (Glutathione S-transferase D1-GFP :gstD1-GFP, a fly construct with ROS-inducible GST promoter upstream to GFP) [31], revealed that compared to Rel, gstD1-GFP expression is much widespread. Interestingly, the inner core MZ is high in both (yellow arrow), while the progenitors housed outside the inner rim (orange arrow) are low in Rel while high in gstD1-GFP expression (**Fig 2A–2A'''**). The developmentally generated ROS is known to evoke Jun Kinase (JNK) signaling, which in turn elicits Fatty Acid Oxidation (FAO) to orchestrate the differentiation of these progenitors [13]. Thus, ROS is the trigger signal for initiating the differentiation program. Since the loss of Rel leads to the differentiation of hemocyte progenitors, we speculated that this might be due to the generation of higher levels of ROS. Two different reporters were employed to check ROS levels: gstD1-GFP and DHE :a redox-sensitive dye, Dihydroxy Ethidium [12]. Surprisingly, as observed by gstD1-GFP expression (**Fig 2B, 2C** and **2E**) and DHE staining (**Fig 2F–2G' and 2I**), inhibition of Rel function from the hemocyte progenitors did not significantly alter ROS levels.

Although ROS levels were significantly higher in progenitors where Rel function was upregulated (Compare **Fig 2B** with **2D** and **2E, 2H, 2H'** and **2I**), the progenitors failed to differentiate (**Fig 1J**, **1O** and **1K, 1P**). The decline in differentiation upon Rel overexpression might be because the ROS levels in the abundant progenitors are insufficient to elicit their differentiation. We, therefore, genetically increased ROS levels by downregulating the ROS scavenger enzyme catalase *(cat)* in conjunction with Rel overexpression to verify this hypothesis. As reported earlier, the enhanced levels of ROS due to the loss of catalase alone can induce ectopic differentiation in the progenitors due to the accumulation of ROS [12]. However, loss of *cat*, in conjunction with Rel overexpression, was unable to rescue the halt in differentiation that is otherwise observed upon Rel overexpression (P1, **Figs 2J–2M, 2N and 2O** and Hnt, **S2I–S2L, S2M**). Therefore, in this genotype, progenitors still experience a halt in their differentiation despite having a high differentiation signal: ROS. These results indicate that Rel has an inhibitory role on the differentiation signal independent of the physiological ROS.

To gain further insight into the reasons for the differentiation defects encountered upon Rel loss, we assayed the status of the signals downstream to ROS, c-Jun N-terminal kinases

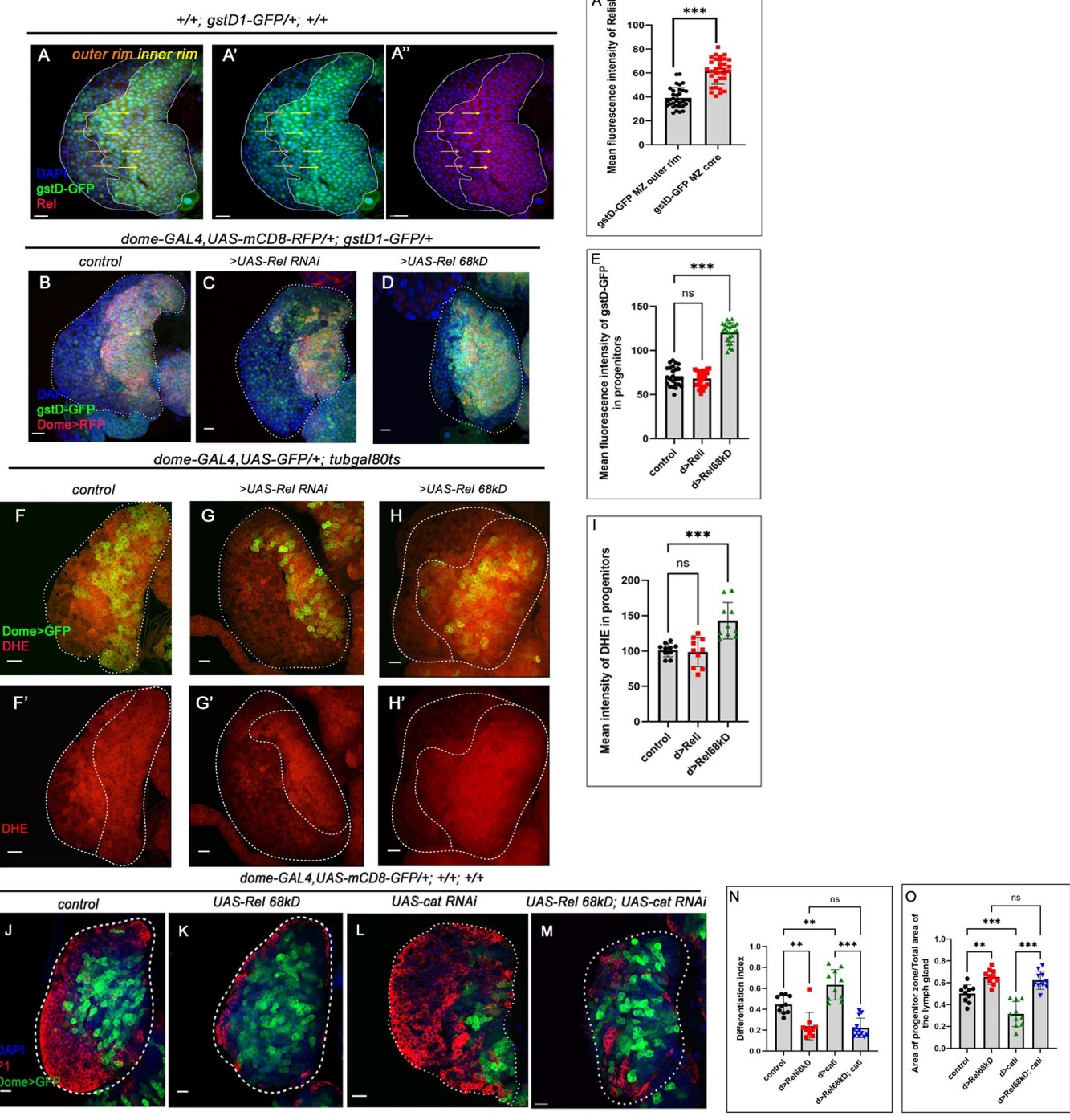

**Fig 2. Progenitor-specific Rel loss leads to ectopic differentiation, which is ROS-independent.** (A-A") Rel expression is enriched in the bulk of the gstD1-GFP expressing progenitors (yellow arrows), while downregulated in the ones present in the peripheral region of the MZ (orange arrows). (A"') Mean fluorescence intensity of Relish expression in the progenitors A-A" (lymph glands n = 33 for mentioned genotype, P-value < .001). (B-D) ROS level in the progenitor visualized by gstD-GFP in control (B) is comparable to Rel loss (C), while significantly high GFP expression can be seen upon progenitor-specific Rel overexpression (D). (E) Mean fluorescence intensity of gstD-GFP expression in the progenitors B-D (lymph glands n≥24 for each genotype, P-values from left to right = 0.615, and <0.001, respectively). (F-H') ROS level (DHE) is high in the progenitors upon sustained Rel expression (H, H'), whereas the levels upon Rel loss from progenitors (G, G') is comparable to control (F, F'). (I) The mean intensity of DHE in the progenitors F-H' (lymph glands n = 10 for each genotype, P-values from left to right = 0.941 and <0.001, respectively). (J-M) UAS-cat RNAi leads to ectopic differentiation (L) while sustained expression of Rel causes halt in differentiation (K) compared to control (J). The co-expression of UAS-cat RNAi and UAS-Rel 68kD (M) is unable to rescue the differentiation defect that is otherwise observed in Rel overexpression (K). (N) Differentiation index for genotype J-M (lymph glands n ≥ 10 for each genotype,

P-values from left to right = 0.001, = 0.004, = 0.988, and <0.001 respectively). (O) The ratio of the area of progenitors to that of the total area of the primary lobe for the genotypes J-M (lymph glands n = 10 for each genotype, P-values from left to right = 0.003, <0.001, = 0.871, and <0.001 respectively). Genotypes are as mentioned. Each dot in the graph represents individual values, Data is expressed as mean ± SD. Statistical analysis: Tukey's multiple comparisons tests. Unpaired t-test with Welch's correction for A'''. P-Value of <0.05, <0.01 and <0.001, mentioned as *, **, *** respectively. Scale bar: 20μ, DAPI: nucleus.

(JNK) in the progenitors. The transcript level of JNK target puckered: *puc* [32] was estimated from FACS-sorted pure progenitor cells from tepIV.Gal4>UAS-2XEGFP that lacked or had sustained Relish expression. Our analysis revealed that loss of Rel function led to a 3-fold (300%) increase in *puc* expression. In contrast, overexpression of Rel resulted in a 0.76-fold (76%) decrease in *puc* expression (**Fig 3A and 3B**), unravelling the activation of JNK downstream to Rel perturbation.

Interestingly, loss of *basket* (*bsk*, *Drosophila* orthologue of Jun-Kinase) from the progenitors is known to prevent their differentiation [12,13], similar to what we observe in Rel overexpression (*Compare* **Figs 3E** and **1O**). Based on these results and our q-PCR analysis, we next performed genetic disruption of JNK activity by expressing a dominant-negative allele of JNK: *UAS-bskDN* from the progenitors that lacked Rel. The excessive differentiation observed upon Rel loss from the progenitor (**Fig 3D**) was rescued upon co-expression of *bsk*DN (**Fig 3F and 3G**). Consequently, the progenitor number was also restored in this genotype compared to Rel loss (**Fig 3C–3F** and **3H**).

FOXO (Forkhead box O) is one of the transcription factors that execute the effects of JNK activation [33,34]. Moreover, studies have shown that FOXO can instigate differentiation downstream of JNK activation [12,13]. In sync with the above studies, our results demonstrated that the halt in differentiation seen upon sustained activation of Rel in the progenitors can be lifted by ectopic expression of FOXO (**S3A–S3D** and **S3E–S3F Fig**). This set of experiments led us to infer that Rel-mediated inhibition of JNK activation failed to evoke the downstream differentiation signals like FOXO. Moreover, the enrichment of Rel expression in the progenitors (Ance+, Pxn- **Fig 1B–1E'**, CHIZ **1S–1T**) and its decline in differentiated blood cells implies that ROS (**Fig 2A–2A'''**) during development cannot elicit differentiation due to the Rel-mediated inhibition on JNK.

Our next goal was to investigate the link between Rel and JNK. In *Drosophila*, Mammalian MAP3 kinase homolog, TAK1 interweaves two pathways that lead to Relish activation and JNK signaling [35–37]. Following immune stimulation, TAK1 activates both the JNK and NF-κB pathways [38–40]. It has also been elucidated that Relish, once activated upon bacterial infection, leads to proteasomal degradation of TAK1, thereby limiting JNK signaling to prevent hyper-immune activation [41].

Interestingly, this pathway also maintains the functionality of the hematopoietic niche during development [21]. Since we observed the activation of JNK in the absence of Relish in the progenitors, we speculated that progenitors might employ the same developmental circuit to prevent their exhaustion. To test our notion, we performed a progenitor-specific genetic loss of TAK1, resulting in a small lymph gland with fewer differentiating cells (Compare **Fig 3I** with **3K** and **3Q**) and abundant progenitors (Compare **Fig 3M** with **3O** and **3R**). This observation suggests that the absence of TAK1 inhibits the differentiation program, which was otherwise initiated by the metabolic signal ROS. Strikingly, simultaneous down-regulation of TAK1 and Rel from the progenitors rescues the excessive differentiation (Compare **Fig 3J**, with **3L**, and **3Q**) and restores the progenitors that are otherwise affected upon Rel loss (Compare **Fig 3N**, with **3P**, and **3R**).

Based on these findings, we can infer that Relish in the MZ prevents precocious JNK activation in the progenitors via inhibiting *tak1* function during development. This regulation, thus,

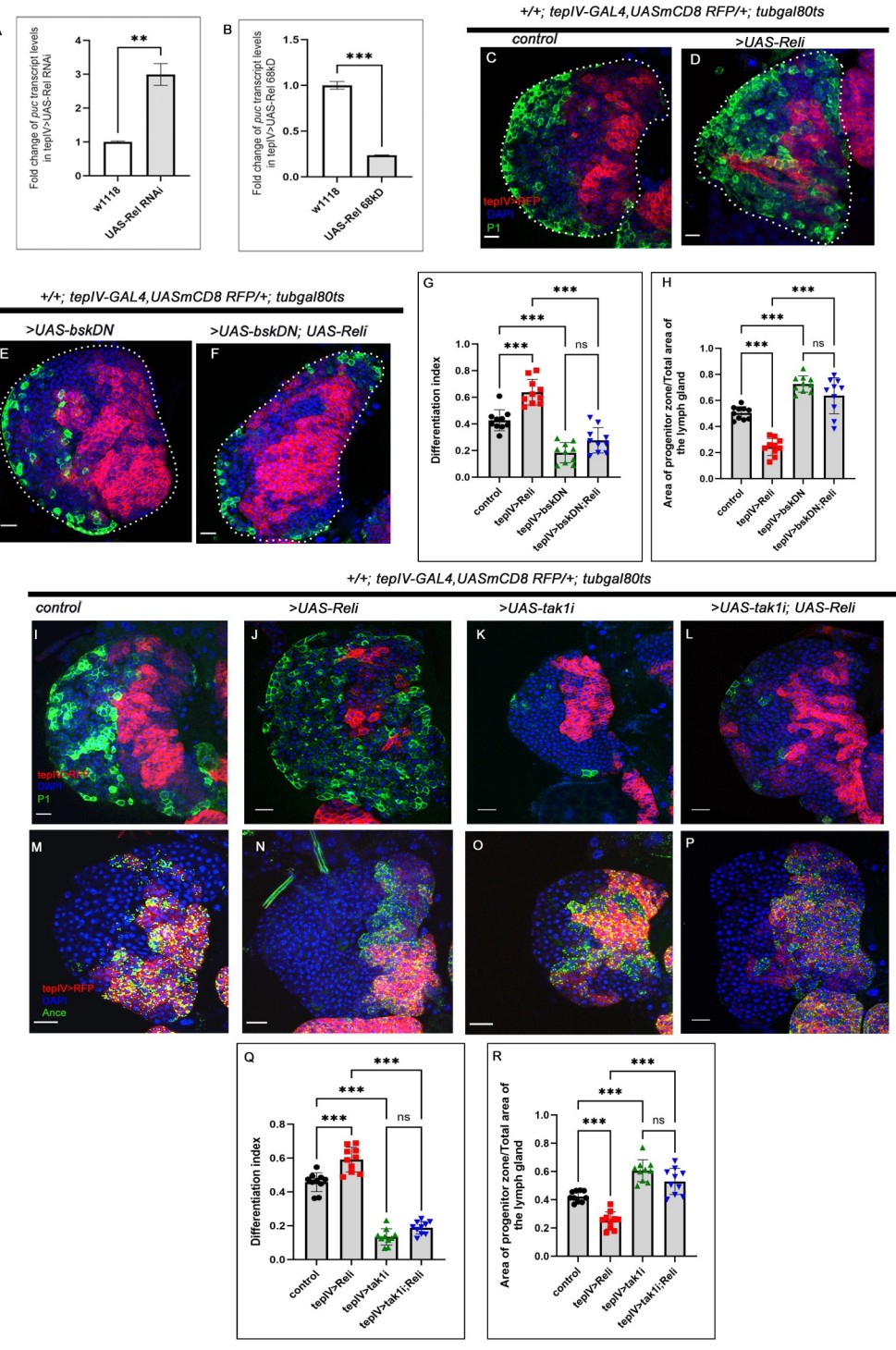

**Fig 3. Relish checks JNK activation in the progenitors via tak1 function to prevent precocious differentiation.**
(A-B) Levels of *puc* transcripts upon progenitor-specific Rel downregulation (A) (N = 3, P-value <0.0083), (B) Rel overexpression causes a decrease in *puc* transcripts (N = 3, P-value: 0.001). The RNA was obtained from FACS-sorted progenitors via GFP expression using *tepIV-GAL4>UAS-2XEGF*. (C-F) Compared to control (C), the loss of JNK activity alone halts the differentiation program (E). Downregulating JNK (by UAS-*bsk*DN) in Rel loss genetic background rescues the ectopic differentiation (F), which was otherwise observed upon Rel loss (D). (G) Differentiation index for genotypes C-F (lymph glands n = 10 for each genotype, P-values from left to right <0.001, <0.001, <0.001, and, = 0.096 respectively). (H) The ratio of the area of progenitors to that of the total area of the primary lobe for the genotypes mentioned in C-F (lymph glands n = 10 for each genotype, P-values from left to right

<0.001, <0.001, <0.001 and = 0.114 respectively). (I-L) Compared to control (I), downregulating TAK1 in Rel loss genetic background rescues the differentiation defects (L), which were otherwise observed upon progenitor-specific Rel loss (J). Loss of *tak1* activity alone halts the differentiation program (K). (M-P) Compared to control (M), a significant increment in progenitor number is evidenced upon loss of *tak1* (O). Downregulating TAK1 in Rel loss genetic background restores the progenitor number (P), which is otherwise declines upon Rel loss (N). (Q) Differentiation index for genotypes I-L (lymph glands n = 10 for each genotype, P-values from left to right <0.001, <0.001, <0.001 and = 0.147 respectively). (R) The ratio of the area of progenitors to that of the total area of the primary lobe for the genotypes mentioned in M-P (lymph glands n = 10 for each genotype, P-values from left to right <0.001, <0.001, <0.001 and = 0.088 respectively). Genotypes are as mentioned. Each dot in the graph represents individual values, Data is expresses as mean±SD. Statistical analysis: Unpaired t-test with Welch's correction for qPCR and rest Tukey's multiple comparisons tests. P-Value of <0.05, <0.01 and<0.001, mentioned as *, **, *** respectively. Scale bar: 20μ, DAPI: nucleus.

restrains the differentiation process evoked upon JNK activation and tilts the balance towards the maintenance program.

## Rel maintains the progenitor pool by inhibiting the JNK-FAO axis

Fatty acid oxidation (FAO) mainly occurs in the mitochondria, and the process involves a series of reactions ultimately resulting in the conversion of FA, which are the building blocks of lipids to acetyl-coenzyme A (acetyl-CoA). The more active FAO is, the more lipids in the system will be mobilized and utilized for energy purposes and vice versa [42,43]. Interestingly, in the Rel loss of function, the residual progenitors have scanty lipid droplets (BODIPY) (**S3G–S3H' and S3J Fig**). In contrast, upon over-expression of Rel, progenitors have more lipid droplets (**S3G–S3G', S3I–S3I' and S3J Fig**) than the control. These observations indicated a possible defect in fatty acid oxidation, which has been linked downstream to JNK facilitating progenitor differentiation [13].

Since Rel loss of function and gain of function alter JNK activity, we decided to check the status of the rate-limiting enzyme of FAO *withered* (*whd*, *Drosophila* homolog of CPT1: Carnitine palmitoyl transferase 1), in lymph gland progenitors in the above-mentioned genetic backgrounds. Loss of Rel in the progenitors resulted in a 2-fold (200%) increase, whereas overexpression of Rel led to a 0.61-fold (61%) reduction in the *whd* transcript level (**Fig 4A and 4B**). Although *Hexokinase A* (*hexA*) transcript level remains unaltered, *Pyruvate kinase* (*pyk*) demonstrates a 0.46-fold (46%) reduction upon Rel loss from the progenitors (**S4C and S4D Fig**).

However, progenitor-specific Rel overexpression led to a 0.6-fold (60%) and a 0.28-fold (28%) increase in the transcript levels of *hexA* and *pyk*, respectively (**S4A and S4B Fig**). We infer that the decrease in *whd* transcripts and increase of *pyk* and *hexA* levels upon sustained Rel expression in the progenitors coaxed them to adopt high-glucose utilization/metabolism while loss of Rel facilitated their differentiation by increasing FAO. Collectively, these experiments further endorsed the involvement of FAO downstream of Rel function on the blood progenitors of the larval lymph gland.

As mentioned earlier, FAO acts downstream of JNK pathway to drive the differentiation program [13]. Our earlier results clearly showed that modulation of Rel expression regulates FAO activity in the progenitors (**Fig 4A and 4B**). So, the ectopic differentiation observed in Rel loss from the progenitors can result from increased FAO. These results indicate that the ectopic differentiation observed in Rel loss might be rescued by blocking FAO. Indeed, the *whd* mutant, when brought in the background of Rel loss from the progenitor, significantly rescued the ectopic differentiation, which is otherwise observed upon loss of Rel (plasmatocytes, **Fig 4C–4F and 4G**). This was further evidenced when simultaneous downregulation of *whd* and *Rel* function from progenitors also yielded a similar response with crystal cells (**S4E–S4H and S4I Fig**). The rescue in the ectopic differentiation was accompanied by the restoration of progenitor area, which is otherwise scanty upon Rel loss (**Fig 4C–4F and 4H**).

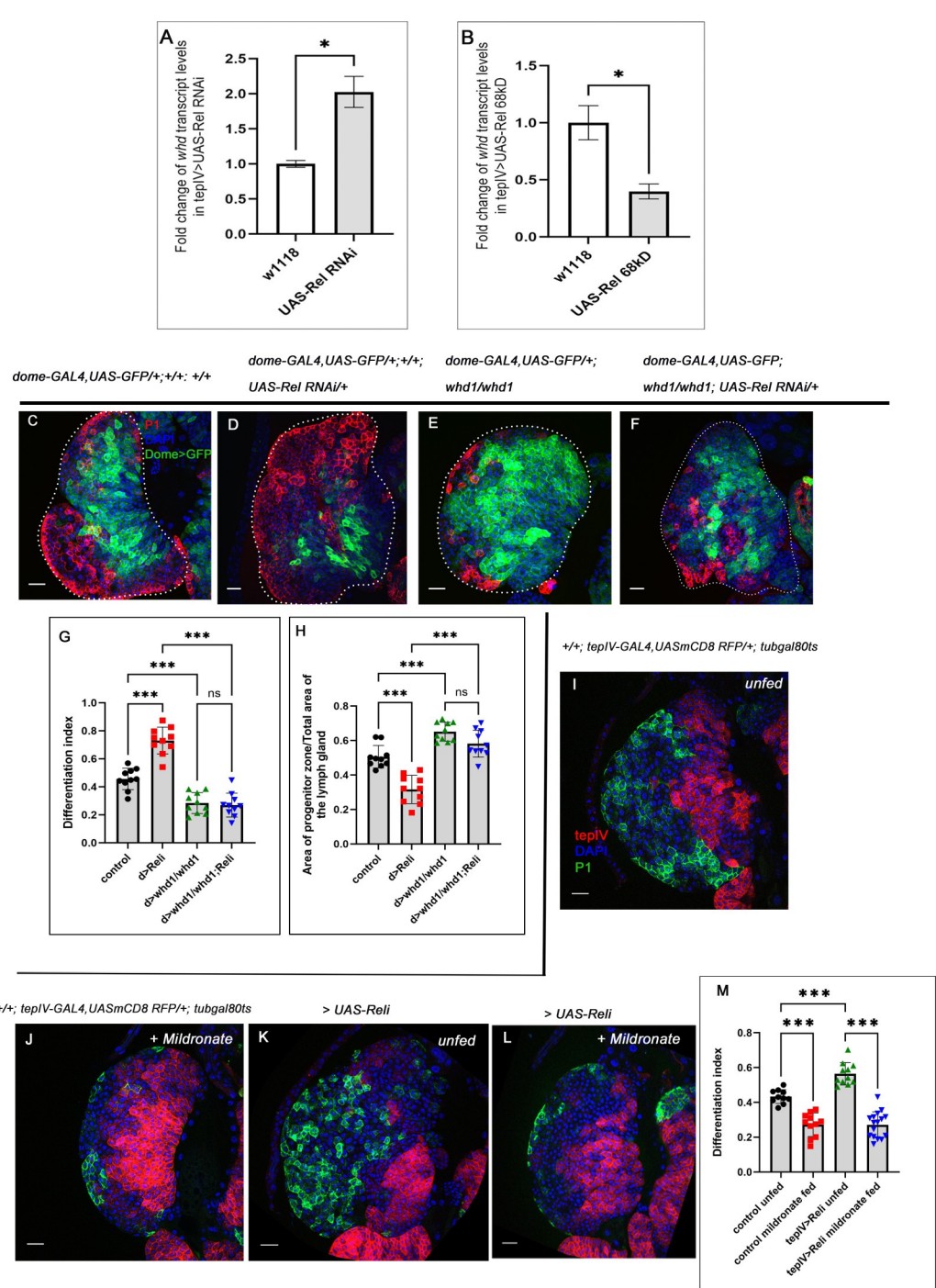

**Fig 4. Rel-mediated inhibition of the JNK-FAO axis is essential for progenitor maintenance.** (A-B) Increase in *whd* transcripts level upon progenitor-specific Rel downregulation (A) (N = 3, P-values: = 0.0121, whereas overexpression (B) caused a decrease in *whd* level (N = 3, P-value: = 0.0101). The RNA was obtained from FACS-sorted progenitors via GFP expression using *tepIV-GAL4>UAS-2XEGFP*. (C-F) Compared to control (C), downregulating FAO using whd null allele (*whd1*) in Rel loss genetic background rescues the excessive differentiation (F), which was otherwise observed in Rel loss from progenitors (D). Downregulating FAO alone results in a halt in progenitor differentiation (E). (G) Differentiation index for genotypes C-F (lymph glands n = 10 for each genotype, P-values from left to right <0.001, <0.001, <0.001 and = 0.979 respectively). (H) The ratio of the area of progenitors to that of the total area of the primary lobe for the genotypes mentioned in C-F (lymph gland n = 10 for each genotype, P-values = <0.001, <0.001, <0.001 and = 0.147 respectively). (I-L) Blocking FAO by mildronate-supplementation in both control (J) and progenitor-specific Rel loss background (L) leads to a decline in differentiation, which is otherwise significantly increased upon progenitor-

specific Rel loss (K) compared to unfed control (I). (M) Differentiation index for genotypes I-L (lymph glands n≥10 for each genotype, P-values from left to right <0.001, <0.001 and <0.001, respectively). Genotypes are as mentioned. Each dot in the graph represents individual values, Data represented as mean±SD. Statistical analysis: Unpaired t-test with Welch's correction for qPCR and rest Tukey's multiple comparisons tests. P-Value of <0.05, <0.01 and<0.001, mentioned as *, **, *** respectively. Scale bar: 20μ, DAPI: nucleus.

To further endorse the genetic data, we carried out pharmacological manipulation of FAO by Mildronate (inhibitor of carnitine biosynthesis and transport) [13,44] and L-Carnitine (increases FAO by enabling the entry of palmitic acid into mitochondria facilitating beta-oxidation) [13,45,46]. Mildronate supplementation rescued the precocious differentiation observed upon Rel loss from progenitors (**Fig 4I–4L** and **4M**)**.** Conversely, L-Carnitine treatment facilitated the progenitors to overcome the halt in differentiation that they experience upon Rel overexpression (**S4J-S4M** and **S4N Fig**).

Based on our expression, genetic and pharmacological studies, we can infer that to maintain a stockpile of blood progenitors and to prevent their differentiation in one go, Rel-mediated developmental regulation on the JNK-FAO axis is essential.

## Rel through inhibition of the JNK-FAO axis modifies the acetylation pattern of histones in progenitor cells

Several reports ascertain that Acetyl CoA is involved in the acetylation of histones [47,48]. In the *Drosophila* lymph gland, loss of FAO in the progenitor cells resulted in compromised H3K9 acetylation, which affected their differentiation to mature blood cells [13]. Since progenitor-specific overexpression of Rel results in the downregulation of the JNK-FAO axis, we hypothesized that the halted differentiation evidenced in such a genotype might be due to an alteration in the acetylation profile of the histones. Compared to the control, a significant reduction in H3K9 acetylation was observed in the Rel overexpression scenario (Compare **Fig 5A–5A" and 5C–5C" and 5F**). In contrast, compared to the control, there was a slight increase in H3K9 acetylation levels in the progenitors upon loss of Rel (Compare **Fig 5A–5A" and 5D–5D" and 5F**). The increase in H3K9 acetylation labeling was further evidenced in mosaic clones generated using *hsflp; act-GAL4* mediated down regulation of Rel. The clonal patches positively marked with GFP (where Rel has been downregulated) show a significant increase in H3K9 acetylation levels *(***Fig 6A–6B'"** and **6D***)* compared to surrounding hemocyte progenitors. While mosaic clones in which Rel was overexpressed exhibited significant decline in H3K9 acetylation levels (**Fig 6A–6A'"** with **6C–6C'"** and **6D**). This alteration in acetylation levels, coupled with the increase or decrease in expression of the components of the JNK-FAO, re-endorsed the antagonistic relationship between Rel and differentiation.

Histone acetylation in eukaryotes majorly relies on acetyl-Coenzyme A (acetyl-CoA). Previous studies have shown that compromised histone acetylation levels can be restored by acetate supplementation [13,49,50]. Hence, the larvae were reared in acetate-supplemented food to ascertain whether the acetylation defect in the *UAS-Rel 68kD* condition halts the differentiation program. Acetate-supplementation led to a dramatic rescue in the acetylation levels in *UAS-Rel 68kD* progenitors compared to the sibling controls reared in normal food (Compare **Fig 5C–5C" and 5E–5E", 5F**). To gain a functional insight into the above observation, we checked the differentiation levels in the same genetic background in control and acetate-supplemented conditions (**Fig 5A and 5B"***)*. In addition to the rescue in acetylation levels, the halt in differentiation that the progenitors experience upon Rel overexpression was relieved with acetate supplementation (plasmatocytes: **Fig 5G–5J** and **5K,** crystal cells: **Fig 6E–6H and 6I**) with a concomitant decrease in progenitor area (**Fig 5G–5J and 5L**).

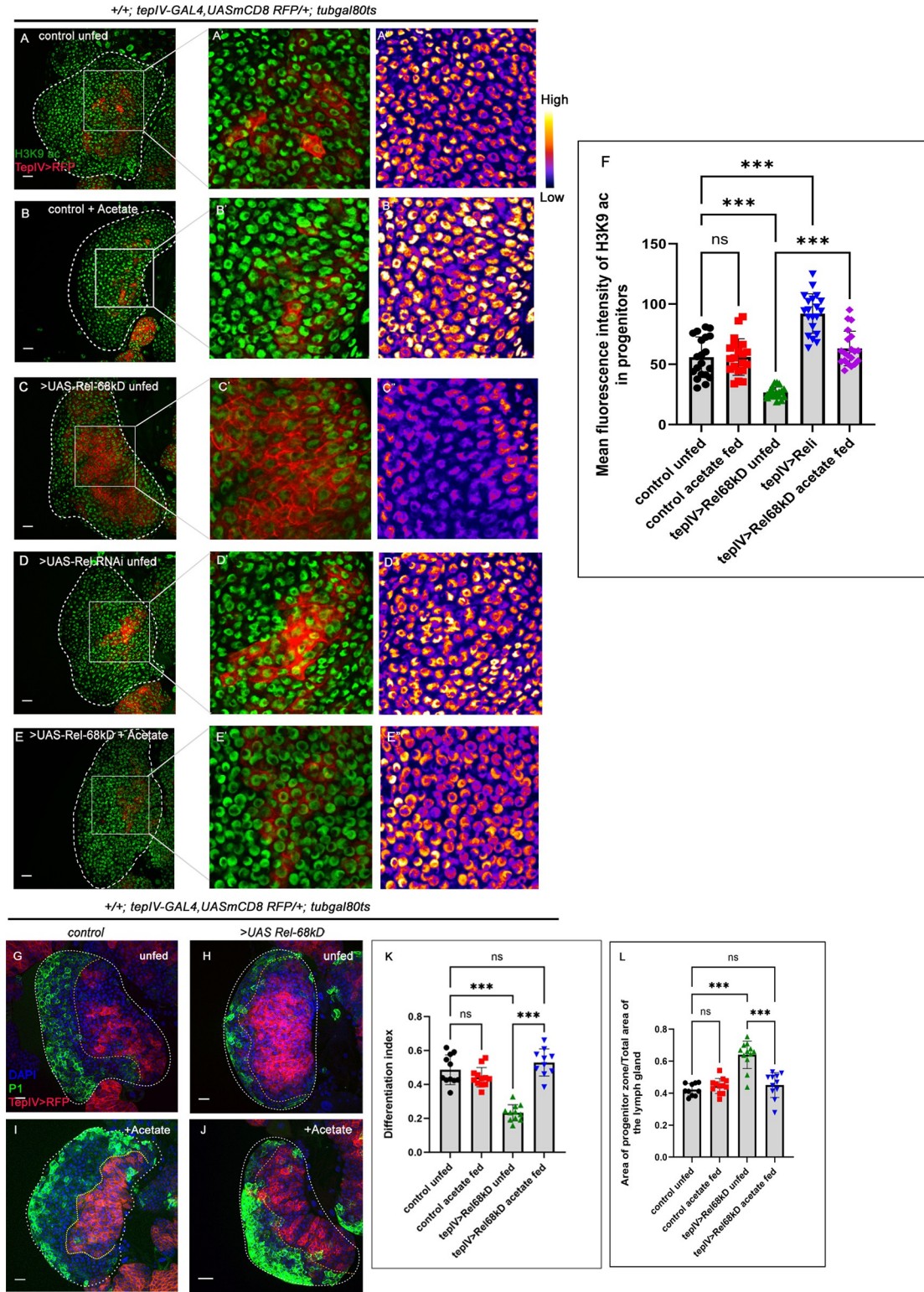

**Fig 5. Upregulated JNK-FAO axis modulates histone acetylation pattern in the progenitors upon Rel loss.** (A-E") Compared to control (A-A"), a drastic reduction in H3K9 acetylation (green) in tep-IV positive progenitors was observed in Rel overexpression (C-C"). In contrast, Rel loss resulted in a slight increase in H3K9 acetylation (D-D"). Acetate-supplementation rescued the acetylation defects seen in the progenitors upon Rel overexpression (E-E") but remained unaltered in control acetate fed (B-B"). (F) The mean fluorescence intensity of H3K9 acetylation in progenitors (A-E") (lymph glands n = 20 for each genotype, P-values from

left to right = 0.999, <0.001, <0.001, and <0.001, respectively). (G-J) Rearing the larvae in acetate-supplemented food rescued the halt in differentiation (J) in the Rel overexpression background, which was otherwise observed in progenitor-specific Rel overexpression (H) and is now comparable to that of control (G). Acetate supplementation in control larvae does not affect differentiation level (I). (K) Differentiation index for genotype in (G-J) (lymph glands n≥10 for each genotype, P-values from left to right = 0.436, <0.001, = 0.489 and <0.001 respectively). (L) The ratio of the area of progenitors to that of the total area of the primary lobe for the genotypes mentioned in G-J (lymph gland n≥10 for each genotype, P-values from left to right = 0.847, <0.001, = 0.766 and <0.001 respectively). Genotypes are as mentioned. Each dot in the graph represents individual values, Data is represented as mean ± SD. Statistical analysis: Tukey's multiple comparisons tests. P-Value of <0.05, <0.01 and<0.001, mentioned as *, **, *** respectively. Scale bar: 20μ, DAPI: nucleus.

Collectively, these results elucidate how, during development, Rel through inhibition of TAK1 antagonizes JNK expression in the progenitor pool, preventing FAO-mediated histone acetylation and thereby controlling the recruitment of progenitors towards the differentiation program.

## Discussion

*Drosophila* third-instar lymph gland houses a pool of primed progenitors arrested in the G2-M phase of the cell cycle. It has been evidenced ROS primes the progenitors for differentiation (Fig 7A–7A'). Interestingly, long before the G2-M arrest sets in, ROS can be detected in the developing progenitors (Fig 7B–7D). Earlier work has demonstrated that the ROS act as the differentiation signal to elicit differentiation by activating JNK-FAO axis [13]. Even though ROS is present in the entire progenitor pool at the late third instar stage (Fig 7E–7E'), differentiation initiates in the outer rim of the medullary zone. This phenomenon spells out the existence of an intrinsic control that overrides the differentiation program in the most of the ROS-enriched progenitors. Our study reveals that *Drosophila* NF-κB factor Relish expresses strongly in the ROS-enriched progenitors and wanes as the cells move towards differentiation during development (Fig 7E'-7E").

We further show that genetic loss of Relish from the progenitors lead to the activation of the JNK-FAO cascade, coaxing them to differentiate.

On the other hand, sustained expression of the N- terminal Relish (UAS-Rel 68 KD) that is known to translocate into the nucleus [51] resulted in a halt in the differentiation program. We found that ROS, the trigger for JNK activation and differentiation in progenitor cells [12], remained unchanged in the Rel loss scenario. Interestingly, elevating ROS levels in the *UAS-Rel 68 kD* genetic background could not rescue the halt in differentiation observed in the latter. Our work demonstrates that Relish in the ROS-enriched progenitors prevents the activation of JNK, thereby deferring their differentiation.

The observation that Relish is responsible for preventing JNK activation demands a node at which the JNK signaling is sensitive to the negative regulation of Relish. The antagonistic relation between NF-κB and JNK is highly conserved across divergent taxa [52,53]. However, different molecules are proposed to act as the mediators to catalyze this inhibition [53,54]. Previous biochemical analysis in S2 cells [41] and genetic studies in the *Drosophila* hematopoietic niche [21] has pinpointed that Rel inhibits TAK1 to control JNK activation. Our genetic data show that during development, Rel inhibits TAK1 to dampen JNK activation, thereby preventing precocious differentiation of the ROS-enriched myeloid-like progenitors. While Rel expression wanes down in the edges of the MZ, the inhibition on TAK1 is lost. The absence of this restrains on TAK1 results in sustained JNK activation, which impinges on FAO and facilitates progenitor differentiation.

The NF-κB/Relish is a crucial transcription factor in the *Drosophila* immune deficiency (Imd) pathway, activating antimicrobial peptide (AMP) transcription to combat gram-

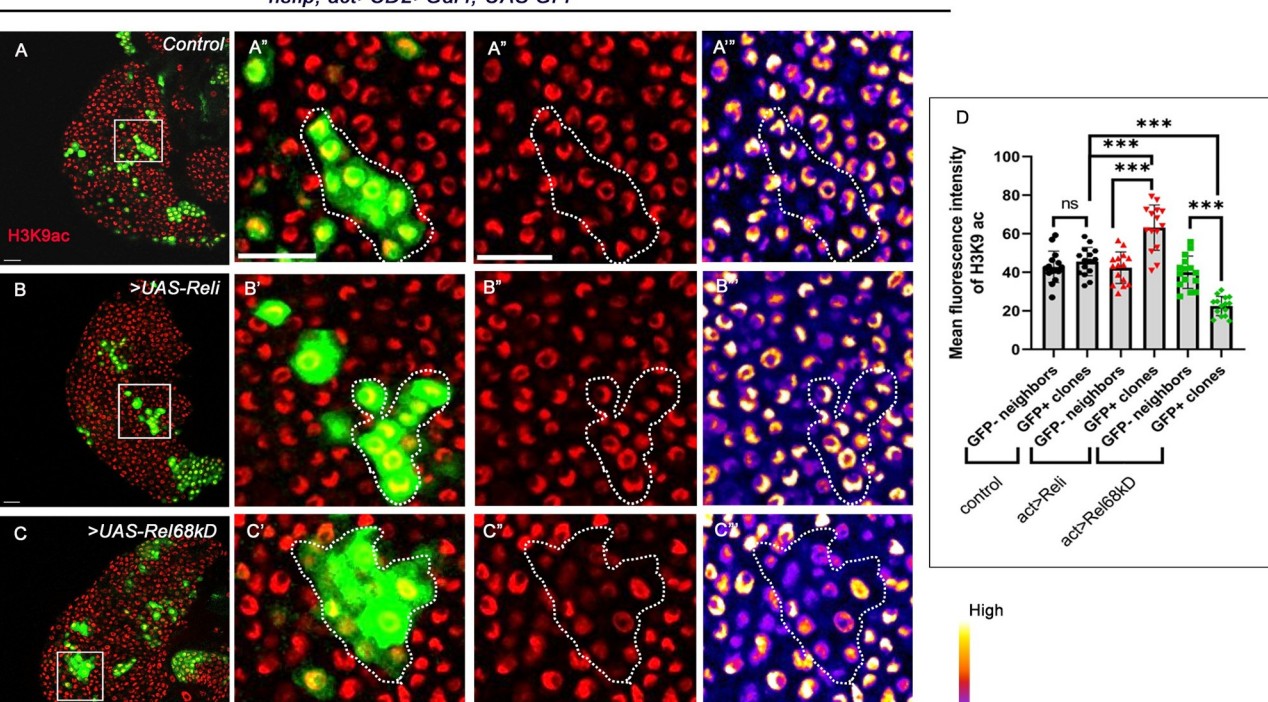

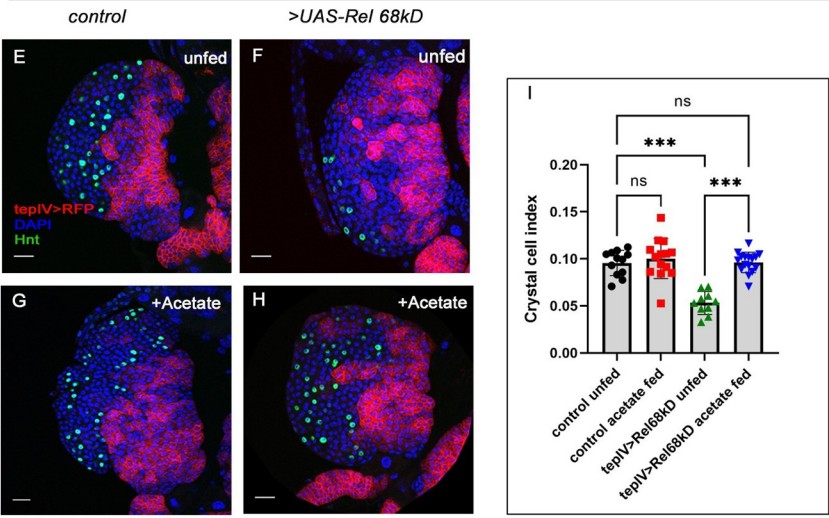

**Fig 6. Clonal analysis reveals that H3K9 acetylation is altered upon perturbation of Rel expression in the progenitor-pool.** (A-C''') Clonal analysis of histone acetylation (H3K9 acetylation) in the GFP-positive *hs-flp; act>CD2>Gal4, UASGFP-based* clonal patches [GFP indicates cells wherein *Rel* function is downregulated (B-B''') or cells with sustained *Rel* expression (C-C''') compared to GFP patches of mock clones (A-A''')]. (D) Quantitative analyses of H3K9 acetylation level in A–C'''. (lymph glands n = 15 for each genotype, P-values from left to right 0.945, < .001, < .001, < .001 and < .001, respectively). (E-H) Feeding the larvae with acetate supplementation rescued the halt in differentiation: crystal cell number (H) in Rel overexpression, which was otherwise observed in progenitor-specific Rel overexpression (F) and is now comparable to that of control (E). Compared to control (E) the number of crystal cells is unaltered in larvae fed on acetate supplemented food (G). (I) Crystal cell index in the genotypes with acetate treatments E-H (lymph gland n≥10, P-values from left to right = 0.861, <0.001, = 0.999 and <0.001, respectively). Genotypes are as mentioned. Each dot in the graph represents individual values, Data expressed as mean ±SD. Statistical analysis: Tukey's multiple comparison tests. P-Value of <0.05, <0.01 and<0.001, mentioned as *, **, *** respectively. Scale bar: 20μ, DAPI: nucleus.

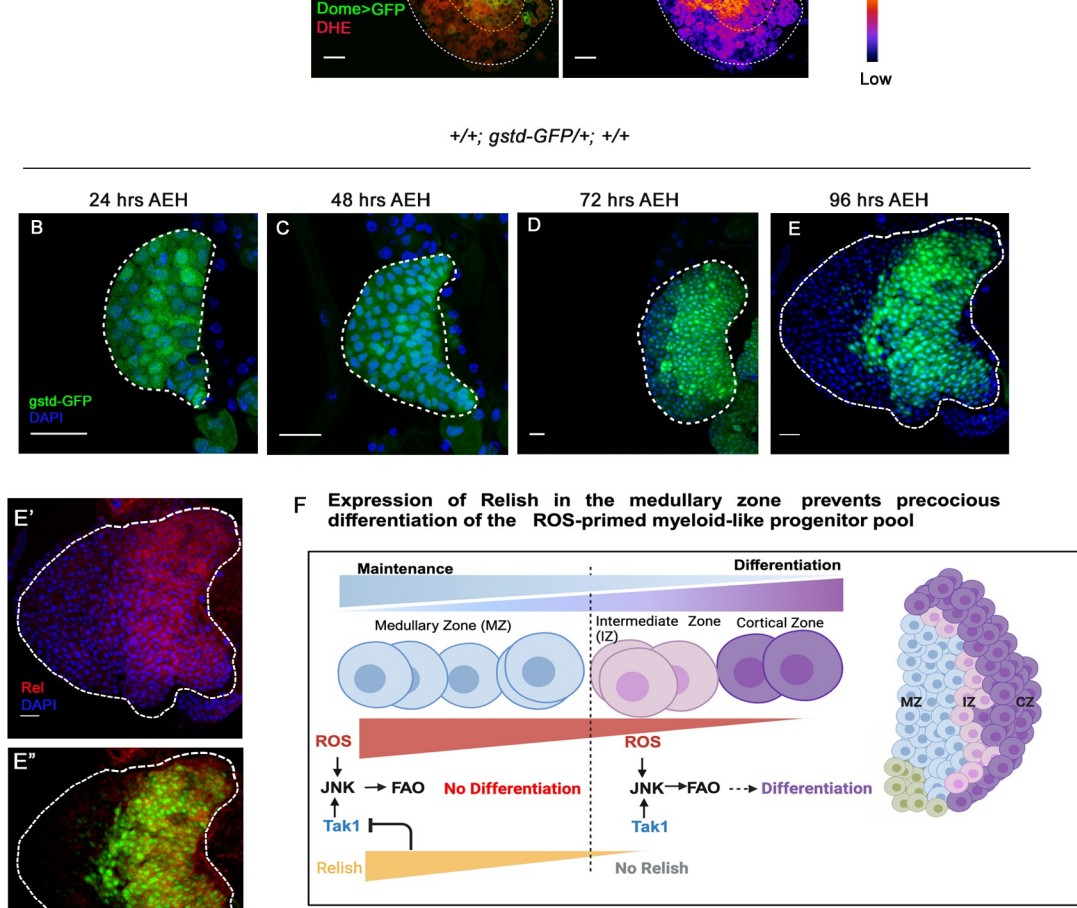

**Fig 7. ROS, the metabolic signal for progenitor differentiation is monitored by the Relish.** (A-A') ROS level visualized by DHE shows high DHE level in the primary lobe progenitors of the late third instar larval lymph gland. (B-E) ROS level visualized by gstD-GFP at different developmental time points 24 hrs (B), 48 hrs (C), 72 hrs (D), 96 hrs (E). (E-E") The bulk of gstD-GFP expressing MZ progenitors are enriched in Rel (F) Schematic representation of our findings depicting how the presence of Rel in the ROS enriched myeloid-like progenitors prevents them from exhaustion.

negative bacterial infections, effectively generating an inflammatory response. Our research has unveiled an intriguing developmental role of Relish, demonstrating its function in maintaining blood progenitors. Employing genetic, molecular and pharmacological studies demonstrate how a metabolic input is calibrated by a developmental signal downstream to Relish to prevent the differentiation program from becoming prodigal (**Fig 7F**). Although the presence of *relish* can be detected in the embryonic lymph gland ([55] Berkeley Drosophila Genome Project, [56,57]) it is intriguing how its expression is restricted to the larval progenitor pool.

Interestingly, our study also demonstrates that overexpressing Rel in progenitor cells can lead to the downregulation of FAO. The negative regulation of NF-κB signaling over Fatty acid oxidation has been reported in various disease conditions. Downregulation in cardiac fatty acid oxidation is a characteristic feature of cardiac hypertrophy. Interestingly, in hypertrophied animals where there was a decline in the level of Fatty Acid Oxidation in cardiomyocytes, administering NF-κB inhibitors showed significant improvement in disease progression [58]. It will be intriguing to explore whether a similar antagonistic relation is active during development in vertebrates.

Downregulation of FAO and subsequent reduction in H3K9 acetylation can be evidenced upon overexpression of Rel in the progenitors compared to the control. Relish (Rel)is also known to govern lipid metabolism during metabolic adaptation in the *Drosophila* fat body. In order to restrain fasting-induced lipolysis and conserve cellular TAG levels, Rel-mediated repression of histone acetylation marks like H3K9 has been reported [59]. Interestingly, rearing *UAS-Rel 68kD* condition in acetate-supplemented food resurrects the acetylation level of H3K9. Studies have shown that acetate supplementation increases histone acetylation by inhibiting HDAC activity and expression [60]. Along with our genetic evidence that illustrates that a decline in H3K9 acetylation upon Rel overexpression is an outcome of compromised FAO, we cannot rule out the possible of involvement of HDACs in this process. In this context, it is important to note that microbe-derived acetate has been shown to activate the *Drosophila* Immunodeficiency (IMD) pathway in a subset of intestinal enteroendocrine cells [61]. However, in our case in sync with previous report, we rule out the involvement of microbe-derived acetate in regulating Rel expression in the lymph gland as no alteration in Rel expression was detected in the lymph glands of axenic larvae [21].

NF-κB regulates vertebrate hematopoiesis through the transcription of genes involved in HSC fate [62] and is known to play a crucial role in HSC emergence from the homogenic endothelium [63]. The process of blood cell development in *Drosophila* shares several similarities with vertebrates [1,2,28,64–66]. Additionally, like *Drosophila*, the vertebrate myeloid progenitors are enriched in ROS [12,67]. It would be interesting to follow up on whether similar crosstalk revealed in our study is operative in vertebrate hematopoiesis.

The cell fate determination of stem and progenitor is vital for normal development and several pathophysiological scenarios. Lymph gland has been exploited to understand the several facets of NF-κB regulation on hematopoietic niches functionality [21,68,69] during infection and development. Interestingly Rel has been shown to evoke mir-317 to fine-tune PGRP-LC expression as well as AMPs in adult flies. This leads to suppressing immune over-activation and restoring immune homeostasis [70]. Apart from cellular signalling pathways and growth factors, cellular metabolism has been majorly implicated in stem and progenitor cell fate determination in recent years. Cell metabolism fuels the bioenergetic requirements and provides essential metabolites that mediate the genetic and epigenetic regulation of cell fate and state. This intricate interaction between cell signalling pathway components and cellular metabolism in determining cell fate is vital for understanding normal development and has implications for disease progression.

In this light, our study foregrounds an essential cog in this wheel by showing how a member of the immune pathway can impinge upon metabolism to regulate progenitor cell fate and homeostasis of tissue during development.

## Materials and methods

### *Drosophila* strains and handling

*Drosophila melanogaster* stocks were maintained at 25°C on the standard medium composed of 9g agar (SRL, Cat# 19661 (0140186)), 18g yeast (KF Instant yeast, India), 51g cornmeal, 45g

white sugar, and 3g Nipagin (Merck, Cat# 61861805001730) dissolved in 20ml absolute Ethanol (Merck, Cat# 1.00983.0511) and 3ml of propionic acid (HIMEDIA, Cat# GRM3658). In general, all crosses were maintained at 25˚C. However, the experiments with RNAi transgenic lines, the culture were maintained at 29˚C.

Synchronized larval batches were prepared for time series experiments mentioned in *Figs 1* and *7*.

Parent flies were starved for 1 hr in an empty vial before transferring them in food vials for 1 hr. The eggs laid were discarded (unsynchronized eggs). Post two such discards, synchronized egg laying was done for 4 hours at 25˚C. The vials with synchronized eggs were kept at respective temperatures as per experimental requirements. For all the experiments, the developmental timeline was determined in terms of hours after egg hatching, AEH [24]. A detailed list of fly strains and genotypes used for the study is provided in S1 Table.

## Clone analysis using FLP-FRT and act-GAL4 system

For the generation of UAS Rel RNAi and UAS Rel68kD clones, we followed the procedure described by Tiwari et al. (2020) [13]. To induce *UAS-rel RNAi* and *UAS-rel 68KD* clones, mid second instar larvae of genotypes: *hsflp; act>CD2>Gal4, UAS GFP; UAS-rel RNAi* and *hsflp; act>CD2>Gal4, UAS GFP, UAS-Rel 68 KD* were subjected to heat shock in food vials for 90 min at 37˚C in a water bath, respectively. Post heat shock, larvae were transferred to 25˚C to recover for 2 hr. Post recovery period larval culture was maintained at 29˚C until dissection for efficient functionality of the GAL4-based driver until dissection at 96 hours AEH.

## Immunohistochemistry

Lymph glands from synchronized batches of larvae were dissected on ice cold 1X PBS. The dissected tissues were fixed in 4% paraformaldehyde (Sigma Aldrich, P6148) prepared in 1X PBS (pH 7.2) for 1 hr at room temperature. Post fixation, two quick washes of 1X PBS were given. Tissues were then permeabilized by 0.3% PBT (0.3 triton-X in 1X PBS) for 30 min (3 washes, 10 min each). Blocking was done in 5% Bovine Serum Albumin [BSA (HiMedia, MB083), prepared in 0.3% PBT] for 45 min-1hr. Tissues were then incubated in primary antibody with appropriate dilution in 5% BSA (prepared in 0.3% PBT) for 18–24 h at 4˚C. Post incubation the tissues were washed by a quick wash of 0.3% PBT followed by three washes of 15 min each with 0.3% PBT.

Tissues were again blocked with 5% BSA for 20–30 min. A secondary antibody specific to the primary antibody was added and kept at 4˚C for 18–24 hr, This was followed by wash using 0.3% PBT (10 min X 3) followed by one quick PBS wash. Post washings, tissues were incubated in DAPI solution for 1 hr at room temperature. After a brief wash in 1X PBS, the samples were mounted in Vectashield (Vector Laboratories, H-1000) and imaged in Zeiss confocal microscopy (LSM 780, and LSM 900). Immunostaining for specific histone acetylation was performed following the previously described protocol [13]. The lymph gland from synchronized larvae was dissected in ice-cold and fixed in 4% PFA prepared in ice-cold 1X PBS (pH 7.2) for 18 hr at 4˚C. Tissues were then permeabilized by 0.3% PBT for 45 min. Blocking was done with 5% BSA made in 1X PBS. Primary antibody and secondary antibody incubation solutions were made in 5% BSA in 1X PBS and washes were done with 0.1% PBT.

Primary antibodies used in this study were: mouse monoclonal anti-Relish (1:50, 21F3), mouse monoclonal anti-β-PS (1:5, CF.6G11), mouse monoclonal anti-Hnt (1:30, 1G9) (Developmental Studies Hybridoma Bank, DHSB), mouse monoclonal anti-P1 (1:100, gift from I. Ando [23]), rabbit monoclonal anti-Ance (1:500, gift from A. D. Shirras [71]), rabbit polyclonal anti-H3K9 acetylation (1:300, 9927, Cell Signaling Technologies). Following secondary antibodies were used for the experiments: mouse polyclonal anti-mouse Cy3 (115-165-166),

donkey polyclonal anti-rabbit Cy3 (711–165–152), goat polyclonal anti-mouse FITC (115–095–166), donkey polyclonal anti-rabbit FITC (711–165–152), donkey polyclonal anti-rabbit 647 (711-605-152). All the secondary antibodies were purchased from Jacksons Immuno Research Laboratories and used as 1:500 dilutions.

## Generation of axenic larva

To obtain axenic larvae, embryos of a particular genotype were transferred to a sterile mesh and cleaned thoroughly with distilled water. Subsequently, following Iatsenko et al. (2018) [72], the embryos were kept in the bleaching solution (Sodium hypochlorite) for 1–2 minutes at room temperature to eliminate the chorion. After discarding the bleaching solution, the embryos were rinsed with autoclaved distilled water.

Thereafter, they were immediately washed with 70% ethanol within the bacterial hood. Inside the bacterial hood, the embryos were delicately moved to a sterile food medium infused with the antibiotic Tetracycline Hydrochloride (10 μg/ml, Sigma Aldrich #T8032) using a sterile brush. Simultaneously, control larvae were prepared by transferring embryos directly to food after washing them with distilled water. Subsequently, all embryos were kept at 29°C until they were dissected at 96 hours after egg hatching (AEH).

## Bacterial plating of axenic larva samples

Approximately five larvae from the control and axenic cohort were sterilized with 70% ethanol. Following the method described in Iatsenko et al. (2018) [72], these larvae were placed in 1.5 ml centrifuge tubes containing 200 μl of sterile LB Broth within a sterile hood for homogenization. The resulting supernatant was mixed with 5 ml of LB Broth and incubated at 29°C overnight to facilitate bacterial growth. Following incubation, the samples were serially diluted and plated onto LB Agar plates. The plates were then incubated for 2–3 days at 29°C until distinct bacterial colonies developed. Images of the bacterial plates were captured using a gel doc (UVP, Bio Spectrum 310 Imaging System).

## Larval infection

Larval infection was done using the *E. coli* DH5-α bacterial strain following the protocol described previously by Ramesh et al. (2021) [21]. Secondary bacterial culture was grown in LB broth at 37°C for 1 hour to the desired OD. The secondary culture was then centrifuged at 5000 rpm for 10 minutes, and the resultant bacterial pellet was resuspended using 1X PBS to our required OD. Four hours before the dissection, the third-instar larvae were washed three times with sterile ddH$_2$O and were gently held still using a paintbrush in a glass slide under the microscope and treated as follows

For sham–A sterilized wounding needle was dipped in 1X PBS and pricked at the position near the posterior end of the larva.

For infection–A needle immersed in bacterial suspension was injected at the aforementioned position of the larva.

Post-injection larvae were again transferred to food plates and kept at 25°C until dissection. The injection was confirmed by looking at the melanized spot on the wounding position while dissecting. All observations were made 4 hr post-infection.

## BODIPY Staining

96 hr AEH larvae were dissected in ice-cold 1X PBS within 15 minutes and immediately fixed with 4% paraformaldehyde at room temperature for 2 hrs. Three PBS washes were given each

for 10 minutes. Samples were then incubated in BODIPY solution (BODIPY493/503, Invitrogen, Cat#D3922; 1:500 dilution prepared in 1X PBS just before use) at room temperature in dark for 1 hr. Subsequently, the samples were washed thrice with 1X PBS for 10 min each. DAPI staining was done for 1 hr at room temperature. This was followed by a single PBS wash for 10 min before mounting in the Vectashield for microscopy. Samples were imaged on the same day or the next day before 24 hr in Zeiss LSM900 confocal microscope. Quantitation was done using ImageJ software. The middle three stacks were merged. The Tep IV positive area was cropped out using a freehand cropping tool and images were saved. BODIPY channel was separated and thresholding was done using Yen thresholding algorithm.

Lipid droplets of size above 4-pixel unit (to eliminate background) were measured using the Analyse Particles tool. Lipid droplet numbers were plotted for all genotypes in GraphPad Prism 9 software.

## Detection of ROS

Larvae were dissected in Schneider's medium (Thermo Scientific, Cat #21720001) followed by incubation in 0.3 μM DHE (Molecular Probes, D11347) in Schneider's medium for 8 min at room temperature in the dark. This step was followed by two washes in 1X PBS for 5 min each; a brief fixation was done with 4% PFA for 10 min. This was followed by two quick 1X PBS washes before mounting in Vectashield and imaged in a Zeiss LSM 900 confocal microscope.

## EdU Incorporation assay

Click-iT EdU (5-ethynyl-2'-deoxyuridine, a thymidine analog) plus kit from Invitrogen (Cat# C10640) was used to perform DNA replication assay. Lymph glands were quickly pulled out in 1X PBS on ice and incubated in EdU solution (1:1000) in 1X PBS on a shaker at room temperature for 30–35 minutes, followed by fixation in 4% paraformaldehyde (prepared in 1XPBS). Post-fixation tissues were permeabilized in 0.3% PBS-Triton for 35 minutes. This was followed by blocking in 10% NGS for 35–40 mins. To detect the incorporated EdU in cells, azide-based fluorophore was used as described in the manufacturer protocol. Next, DAPI was done in 1X PBS and then mounted in Vectashield.

## Metabolic supplements and inhibitors

Post 36-hour AEH, larvae were reared in food supplemented with Fatty acid β-oxidation inhibitor: Mildronate (Cayman Chemicals, Cay15997) or to upregulate FAO by L-carnitine (Sigma-Aldrich, C0283) at a concentration of 100 μM and Sodium acetate (Sigma-Aldrich, 71196) at a concentration of 50 mM [13]. All analyses were done in the late third instar stages. Similarly, aged larvae fed on vehicle controls served as control larvae. Control larvae were reared in fly food supplemented with vehicle control for all feeding experiments.

## Single-cell sorting and RNA isolation

Sorting was performed to obtain a pure population of late-stage progenitors using lymph glands from the *tepIV-GAL4>UAS-2XEGFP* fly line. 70–80 Lymph glands were dissected from third-late instar larvae (96 hr AEH) within 30 minutes in 1X Schneider's medium (Thermo Scientific, Cat #21720001) with 5% Fetal Bovine Serum (FBS) (Thermo Scientific, Cat# 16140071). Tissues were collected in 500μl of above medium and subjected to cell dissociation by replacing the media with 200μl of 10 X TryplE (Gibco Cat # A1217701) solutions, thereafter incubated at 37°C for 30 minutes, followed by vigorous pipetting. The action of TryplE was then neutralized by adding 300μl of 10% FBS prepared in 1X Schneider's medium. The

mediums containing the cells are then passed through Flowmi cell strainer 40μm (Sigma Aldrich, Cat# BAH136800040) to remove unwanted debris or cell clumps, if any. These filtered cells were kept at 4°C and GFP positive cells were sorted immediately using Leica BD FACS Aria Fusion Cell sorter with an excitation laser of 488 nm. GFP-expressing cells were sorted using a 100-micron nozzle at a fluidics pressure of 20 psi immediately by fluorescence-activated cell sorter (BD FACS Aria Fusion). Wild-type non-GFP control cells were used to set the auto fluorescence. Sample acquisition and cell sorting was carried out using BD FACS Diva (Version 8.0.1). A gating strategy for cell sorting was followed by FSc-A Vs SSc-A to gate the cluster of a healthy homogenous cell population, followed by doublet discrimination using FSc-H Vs FSc-W, and SSc-H Vs SSc- W and a dot plot of 530/30 BP (GFP) vs 582/15 (PE) filters were used to eliminate high background signals from dim GFP-expressing cells. A threshold value was set at 15000 empirically to eliminate the detection of irrelevant debris while acquiring data. The sorted cells were then collected in 10% ice-cold FBS prepared in 1X Schneider's medium for RNA isolation. The collected cells were spun down at 4°C under 2500 rcf for 10 minutes. The supernatant was carefully discarded and the cell pellet was resuspended in 200μl of TRIzol (Invitrogen, Cat# 15596018). The cells were then homogenized using a handheld motorized homogenizer. The volume was made up to 500μl and kept in room temperature for 5 minutes to allow complete dissociation of the nucleoprotein complex.

200μl chloroform (Sigma # C2432) was added and mixed thoroughly by vigorous shaking for about 15 seconds and incubated in room temperature for 3 minutes. Phase separation was done at 4°C for 15 minutes under a centrifugal force of 13000 rcf. The transparent aqueous phase containing the RNA was transferred in a new 1.5ml microcentrifuge tube and 0.5μl of glycogen (Sigma Cat# G1767) was added to obtain higher precipitation of RNA. After that the equal volume of isopropanol (2-propanol) (Sigma Cat# I9516) was added and mixed gently. The tube was then kept at -80°C for overnight (about 12 hours). This step was followed by centrifuging at 4°C for 15 minutes (15000 rcf) and the RNA was precipitated in the form of a white gel-like pellet in the bottom of the tube. The supernatant was discarded and the pellet was subjected to two consecutive washes for 10 minutes at 4°C with 75% Ethanol (Sigma-Merck Cat# 1.00983.0511) prepared in Autoclaved Type I distilled water, under 10000 rcf and 7500 rcf respectively. The pellet was further subjected to another dry run to aspirate out any remaining traces of Ethanol and air-dried at room temperature for 10 minutes and resuspended in 20μl of autoclaved Type I distilled water. The quantity and quality of RNA was checked using the Thermo Fisher Scientific Nanodrop One C spectrophotometer and then stored at -20°C until further use.

## cDNA synthesis and RT-qPCR

cDNA was synthesized using Verso cDNA synthesis kit (Thermo Scientific, Cat# AB1453A) following the manufacturer's recommended protocol. Real time-qPCR (RT-qPCR) was conducted using the Bio-Rad CFX96 Touch Real-Time PCR Detection System. Universal SYBR Green Supermix (Bio-Rad, Cat# 172–5124) was used for the RT-qPCR reactions. The relative transcript labels were measured using the 2-ΔCt method. The transcript levels were normalized with respect to constitutively expressing ribosomal protein 49 (rp49) mRNA. Each RT-qPCR experiments were conducted on three technical repeats of three independent biological repeats. Statistical analysis: Unpaired t-test with Welch's correction. Primers spanning exon-exon junction were used, list of the same is available in S2 Table.

## Fluorescence quantification

Confocal z-stacks of dissected tissue were acquired with a 1 μm step and identical laser power and scan settings. Images obtained were analyzed by either ImageJ/Fiji software (NIH). For

determining the Mean Fluorescence of Intensity of *gstD-GFP* or DHE, the mean gray values of the region of interests (ROI) for the desired z stacks were determined using the software. The mean gray value was calculated by subtracting the gray value of the background from the gray value of the desired ROI. For H3K9 intensity analysis single stack was employed from the middle section of the Z-stacks. Data is expressed as mean+/- Standard Deviation of values and are plotted in GraphPad prism. All statistical analyses were performed employing Tukey's multiple comparison.

## Quantification for EdU and FUCCI

Counting the number of EdU+ and FUCCI+ progenitors in lymph glands was done as described earlier [11], using spot detection tool in Imaris (Bitplane) software and normalized by total number of nuclei in the progenitor area per primary lobe of lymph gland. For FUCCI analysis, by utilizing the spot detection tool the numbers of red, green and yellow nuclei were counted in Dome+ surface. The number of only red cells and only green cells were calculated by subtracting total number of yellow cells from total number of red cells and green cells respectively. Once this is done, percentage of S, G1 and G2-M phase was calculated by using the simple formula below:

Percentage of S phase = Number of only red cells/Total number of cells in progenitor zone

Percentage of G1 phase = Number of only green cells/Total number of cells in progenitor zone

Percentage of G2-M phase = Number yellow cells/Total number of cells in progenitor zone.

Pie chart was made on the average percentage data

## Differentiation index calculation

The differentiation index was calculated taking the middlemost stacks from confocal z sections were merged into a single stack for each lymph gland lobe using ImageJ/Fiji (NIH) software as described earlier [73]. Using the Free hand tool marked P1 positive area. The size was measured using the Measure tool (Analyse–Measure). Similarly, the DAPI area was also measured. To calculate the differentiation index we divided the size of the P1 positive area by the total size of the lobe (DAPI area). For each genotype, mostly ten lymph gland lobes were used, and statistical analysis was performed using Tukey's multiple comparison test.

For calculating crystal cell index and intermediate progenitor index total number of nuclei (Hoechst or DAPI), Hnt positive cells, double positive cells for Dome and Pxn marker or CHIZ positive cells were counted using Imaris software [13,22]. Once we had the data, the number of Hnt-positive cells and Dome+ and Pxn+ cells or CHIZ positive cells divided by the total number of nuclei for crystal cell index and intermediate progenitor index respectively. The number obtained was further plotted in GraphPad prism 9.

## Quantitation and statistical analysis

In general, the quantitative analysis of data was performed, and graphs were plotted in GraphPad prism 9. Tukey's multiple comparison test was employed to determine statistical analysis and result of the analysis is expressed in graphs in the form of asterisk as follows *p<0.05, **p<0.01, and ***p<0.001. Data represented as mean ± SD. For comparison of two paired data groups using samples RT-qPCR data Unpaired t-test with Welch's correction was done.

## Supporting information

**S1 Fig. Loss of Relish from the progenitors leads to precocious differentiation.** (A-C) Progenitor-specific loss of Rel increased crystal cell index (Hnt, green) (B) whereas overexpression of Rel resulted in a decrease (C) compared to control (A). (D) Differentiation index (crystal cells) for genotypes (A-C) (lymph glands n = 10 for each genotype, P-values from left to right = 0.005, and = 0.016 respectively). (E-F) No lamellocyte induction was observed in Rel loss from the progenitor (β-PS, green). (G) To check the presence of commensal gut microbiota, larval homogenates were spread on LB Agar plates. Compared to control where bacterial colonies were visible post incubation, axenic condition had no growth. (H-I) Differentiation status (*hml>GFP*) in the lymph gland from larvae reared in the axenic condition is comparable to control. (J) Differentiation index for each condition of H-I (lymph gland n≥16, P-value = 0.608). (K-K') Progenitor-specific Relish expression remains unaltered in axenic condition (compare K with K'). (K") The mean fluorescence intensity of Relish in progenitors for each condition of K-K' (lymph gland n≥19, P-value = 0.066). (L-M') Compare to sham (L–L'), a significant reduction in Relish expression (arrow) was observed in the hematopoietic niche 4 hr post-infection (M-M'). (N–P') Compare to uninfected conditions (N–N') and sham (O–O'), no significant change in Relish expression was observed in the hematopoietic progenitors 4 hr post-infection (P-P'). (Q) Statistical analysis of the data in (N–P') (lymph glands n = 15 for each treatment, P-values from left to right = 0.780, = 0.173 and = 0.487, respectively). (R-R") Another progenitor-specific driver, *dome-Gal4* gave similar results like *tepIV-Gal4*. (R"') The ratio of the area of progenitors to that of the total area of the primary lobe for the genotypes R-R" (lymph glands n = 10, P-values from left to right <0.001 and <0.001, respectively). (S-S") A significant increase in terminally differentiated (P1, red) was observed in Rel loss employing *dome-Gal4*, whereas overexpression of Rel resulted in a halt in differentiation compared to control. (S"') Differentiation index for genotypes in S-S" (lymph gland n = 10, P-values from left to right <0.001 and <0.001, respectively). (T-T") Progenitor-specific loss of Rel results in decreased intermediate progenitors (Dome: red and Pxn: green, T'), on the other hand, sustained Rel expression leads to an increase of intermediate progenitors (T") compared to control (T). (T"') The ratio of the number of intermediate progenitors to that of the total number of cells in the primary lobe for the genotypes T-T" (lymph glands n≥12, P-values from left to right <0.001 and <0.001, respectively). Genotypes are as mentioned. Each dot in the graph represents individual values, Data expressed as mean ± SD. Statistical analysis: Tukey's multiple comparison tests. Unpaired t-test with Welch's correction for J and K". P-Value of <0.05, <0.01 and<0.001, mentioned as *, **, *** respectively. Scale bar: 20μ, DAPI: nucleus.
(TIF)

**S2 Fig. Progenitor-specific Rel loss leads to alteration in cell cycle.** (A-C) Both loss of Rel (B) and overexpression of Rel (C) resulted in an increase of S phase (red) and G1 phase (green) at the expense of G2-M (yellow) arrested cells otherwise found in control lymph gland progenitors (A). (D-D") Infographic representation of results in A-C. (E-G) EdU incorporation assay endorsed the findings of A-C. (H) Rate of EdU incorporation in genotypes E-G (lymph glands n≥12, P-values from left to right <0.03 for control versus UAS-Reli and <0.001 for control versus UAS-Rel68kD). (I-L) Compared to control (I), *UAS-cat RNAi* expression in the progenitors increases crystal cell differentiation (K). The co-expression of *UAS-cat RNAi* and *UAS-Rel 68kD* (L) is unable to rescue the halt in differentiation (crystal cells), which is observed in Rel overexpression (J). (M) Differentiation index (crystal cells) for genotypes I-L (lymph glands n≥12, P-values from left to right <0.001, <0.001, = 0.298 and <0.001). Genotypes are as mentioned. Each dot in the graph represents individual values, Data expressed as mean ±SD.

Statistical analysis: Tukey's multiple comparison tests. P-Value of <0.05, <0.01 and<0.001, mentioned as *, **, *** respectively. Scale bar: 20μ, DAPI: nucleus.
(TIF)

**S3 Fig. FOXO overexpression can rescue differentiation defects upon Rel overexpression.**
(A-D) Compared to control (A), co-expression of FOXO and Rel 68 kD rescues the halt in differentiation (D), which was otherwise observed in Rel overexpression scenario (C). Upregulating FOXO activity alone resulted in ectopic differentiation (B). (E) Differentiation index for genotypes A-D (lymph glands n = 10, P-values from left to right <0.001, <0.001, <0.001 and <0.001, for UAS-Rel68kD versus UAS-Rel 68kD; UAS-FOXO). (F) The ratio of the area of progenitors to that of the total area of the primary lobe for the genotypes A-D (lymph glands n = 10, P-values from left to right <0.001, <0.001, <0.001 and <0.001, respectively). (G-I') Neutral lipid content visualized by BODIPY in the progenitors, compared to control lymph glands (G-G') a drastic reduction in lipid accumulation was observed in the Rel loss condition (H-H'), whereas Rel overexpression increased the number of lipid droplets (I-I'). (J) Total number of lipid droplets in the progenitors G-I' (lymph glands n≥17, P-values. from left to right <0.001 and <0 .005 respectively). Genotypes are as mentioned. Each dot in the graph represents individual values, Data expressed as mean ± SD. Statistical analysis: Tukey's multiple comparison tests. P- Value of <0.05,<0.01 and<0.001, mentioned as *, **, *** respectively. Scale bar: 20μ, DAPI: nucleus.
(TIF)

**S4 Fig. Rel maintains the progenitor pool by having an inhibitory regulation over the JNK-FAO axis.** (A-D) Rel overexpression causes increments of *hexA* and *pyk* transcript levels respectively (A and B) (N = 3, P-values from left to right = 0.011 and = 0.023, respectively). The level of *hexA* remains unchanged (C) upon progenitor-specific Rel loss, whereas *pyk* transcripts decreases (D) (N = 3, P-values from left to right = 0.0593 and = 0.0175, respectively). The RNA was obtained from FACS-sorted progenitors via GFP expression using *tepIV-GAL4>UAS-2XEGFP*. (E-H) Compared to control (E), downregulating FAO using UAS-*whd* RNAi in Rel loss genetic background rescues the excessive differentiation: crystal cells (H), which was otherwise observed in Rel loss from progenitors (F). Downregulating *whd* alone results in a decline in crystal cell number (G). (I) Differentiation index (crystal cells) for genotypes, (E-H) (lymph glands n≥12, P-values from left to right <0.001, <0.001, <0.001 and = 0.169 respectively). (J-M) Feeding the larvae with L-carnitine supplemented food in both control (K) and progenitor-specific Rel overexpression (M) caused precocious differentiation, which was otherwise not observed in progenitor-specific Rel overexpression (L) and unfed control (J). (N) Differentiation index for genotypes J-M (lymph glands n≥10, P-values from left to right <0.001, <0.001 and <0.001, respectively). Genotypes are as mentioned. Each dot in the graph represents individual values, Data expressed as mean± SD. Statistical analysis: q-PCR: Unpaired t-test with Welch's correction and rest Tukey's multiple comparisons tests. P-Value of <0.05,<0.01 and<0.001, mentioned as *, **, *** respectively. Scale bar: 20μ, DAPI: nucleus.
(TIF)

**S1 Table. Fly Stocks used in the current study.**
(DOCX)

**S2 Table. List of primers used in the current study for RT-qPCR.**
(DOCX)

**S1 Data. The excel file containing the quantitative data used for panels presented in Figs 1–6 and S1–S4.**
(XLSX)

## Acknowledgments

We thank Profs D Bohman, I Ando, U Banerjee, J Fessler and A D Shirras for reagents. We thank IISER Mohali's Confocal Facility and FACS Facility, the Bloomington Drosophila Stock Center at Indiana University, Kyoto Stock Center flies and Developmental Studies Hybridoma Bank, University of Iowa for flies and antibodies. Special thanks to Devki and Pratik Arora for assisting us with cell sorting experiments. Models 'Created with BioRender.com'.

## Author Contributions

**Conceptualization:** Parvathy Ramesh, Satish Kumar Tiwari, Lolitika Mandal.

**Data curation:** Parvathy Ramesh, Satish Kumar Tiwari, Md Kaizer, Kaustuv Ghosh, Lolitika Mandal.

**Formal analysis:** Parvathy Ramesh, Satish Kumar Tiwari, Md Kaizer, Deepak Jangra, Sudip Mandal, Lolitika Mandal.

**Funding acquisition:** Lolitika Mandal.

**Investigation:** Md Kaizer, Deepak Jangra, Kaustuv Ghosh, Sudip Mandal.

**Project administration:** Lolitika Mandal.

**Supervision:** Lolitika Mandal.

**Validation:** Parvathy Ramesh, Satish Kumar Tiwari, Md Kaizer, Deepak Jangra.

**Visualization:** Parvathy Ramesh, Md Kaizer, Deepak Jangra.

**Writing – original draft:** Parvathy Ramesh, Md Kaizer, Lolitika Mandal.

**Writing – review & editing:** Md Kaizer, Sudip Mandal, Lolitika Mandal.

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
