## [Decision Letter · Decision Letter 0]

11 Mar 2024

Dear Dr Mandal,

Thank you very much for submitting your Research Article entitled 'The NF-κB Factor Relish maintains blood progenitor homeostasis in the developing Drosophila lymph gland' to PLOS Genetics.

The manuscript was fully evaluated at the editorial level and by independent peer reviewers. The reviewers appreciated the attention to an important problem, but raised some substantial concerns about the current manuscript. Based on the reviews, we will not be able to accept this version of the manuscript, but we would be willing to review a much-revised version. We cannot, of course, promise publication at that time.

If you decide to revise the manuscript for further consideration at PLOS Genetics, please aim to resubmit within the next 60 days, unless it will take extra time to address the concerns of the reviewers, in which case we would appreciate an expected resubmission date by email to plosgenetics@plos.org.

We are sorry that we cannot be more positive about your manuscript at this stage. Please do not hesitate to contact us if you have any concerns or questions.

Yours sincerely,

Pablo Wappner

Academic Editor

PLOS Genetics

Gregory P. Copenhaver

Editor-in-Chief

PLOS Genetics

Reviewer's Responses to Questions

**Comments to the Authors:**

Reviewer #1: “The NF-κB Factor Relish maintains blood progenitor homeostasis in the developing Drosophila lymph gland” is an elegant genetic study revealing the novel role of Drosophila NF-kB-like factor Relish (Rel) in preventing hematopoietic progenitors from undergoing ROS-induced differentiation. This study addressed a significant question- what is the mechanism that protects the progenitors from entering the differentiation program in the presence of ROS that is known to trigger their differentiation? They showed that Rel loss results in the precocious differentiation of the hematopoietic progenitors, suggesting a protective role. As Rel has a well-documented role in immune response, they have provided evidence to argue that the protective role of Rel is not related to a evoke in the immune response. Importantly, Rel overexpression led to a block in progenitor differentiation, an effect opposite to that caused by Rel loss. In terms of mechanism of action, the authors have convincingly demonstrated that Rel has an inhibitory role towards JNK activation, thereby preventing progenitor differentiation. As fatty acid oxidation (FAO) acts downstream of JNK for progenitor differentiation, they went to show that FAO as well as FAO-mediated histone acetylation also act downstream of Rel in controlling progenitor differentiation program via genetic analysis and pharmacological treatment. Overall, the experiments have been executed meticulously, with proper controls, and the data are of high quality, supported by statistical analyses. Moreover, the manuscript is well-written and a pleasure to read. The findings will be of great interest to general readers of PLOS Genetics.

Major comments:

1. Rel is a transcription factor, a well-known fact that was not mentioned in this manuscript. This needs to be improved. It is unclear how Rel negatively regulates JNK pathway. Although not essential for the publication of this manuscript, the reviewer would like to see a discussion on how Rel might be linked to JNK pathway during hematopoietic progenitor lineage development.

2. It is interesting that there is a region-specific differentiation in the outer rim of the MZ. How is it regulated by Rel? Any spatial regulation of Rel (ie any gradient localization of Rel from inner to outer rim)? Has single-cell RNA sequencing been performed on the lymph gland to support the model in Fig S5K? Again, these are not essential for this manuscript, but will be good to include a discussion to point at future directions.

Minor comments:

1. When mentioning “Lamellocytes”, indicate that they are “a class of hemocytes”.

2. In “These results indicate that the ectopic differentiation observed in Rel loss might be rescued by blocking FAO and Rel simultaneously.”, remove “and Rel simultaneously”.

Reviewer #2: uploaded as attachment

Reviewer #3: The Authors of this manuscript use a complex analysis by combining a genetic approach, as genetic manipulation, marker expression, morphological, gene expressional and pharmacological studies to reveal the factors and their interactions in the regulation of Drosophila blood progenitor cells. They show that Relish, an NF-κB-like factor, a key regulator of antimicrobial defense is a major factor too for preventing the entire progenitor pool from differentiation at one go, acting in a histologically defined region of the central hematopoietic organ. The complex analysis reveals the so far ill-defined role of Relish in regulation of hematopoiesis in normal development.

Although the multifaceted study is noticeably clearly presented and the phenotypic analysis is supported by schemes in several figures, which helps to read, a graphical abstract-like figure showing the integration of the Rel-related activation would be essential in order to help the reader to easily understand and comprehend the integration of this novel phenomenon into the already described regulatory networks.

**Have all data underlying the figures and results presented in the manuscript been provided?**

Reviewer #1: Yes

Reviewer #2: Yes

Reviewer #3: Yes

PLOS authors have the option to publish the peer review history of their article (what does this mean?). If published, this will include your full peer review and any attached files.

Reviewer #1: No

Reviewer #2: No

Reviewer #3: No

---

## [Decision Letter · Decision Letter 1]

17 Jul 2024

Dear Dr Mandal,

Thank you very much for submitting the revised version of your Article 'The NF-κB Factor Relish maintains blood progenitor homeostasis in the developing Drosophila lymph gland' to PLOS Genetics.

The manuscript was evaluated again by the reviewers 1 and 2 of the original submission, and I have evaluated myself the revisions in response to the comments of reviewer 3, which I found properly addressed. While the reviewer 1 found that his/her concerns were fully attended, reviewer 2 still feels that some aspects of the discussion need to be expanded.

We therefore ask you to modify the manuscript according to reviewer 2 recommendations.

In addition we ask that you to:

1) Please, provide a detailed list of your responses to his/her comments and a description of the changes you have made in the manuscript.

To resubmit, log into your Editorial Manager account and select the option 'Revise Submission' in the 'Submissions Needing Revision' folder.

Yours sincerely,

Pablo Wappner

Section Editor

PLOS Genetics

Gregory Copenhaver

Section Editor

PLOS Genetics

Reviewer's Responses to Questions

**Comments to the Authors:**

Reviewer #1: The authors have adequately addressed all my concerns. It is ready for publication in PLOS Genetics.

Reviewer #2: This reviewer appreciates the thought and diligence the authors have put into this revision, particularly the clonal analysis, and the new model diagram.

In my opinion, however, the significance of this contribution would be greatly enhanced EITHER

(1) by some understanding of HOW the Relish restricted expression domain is generated, including whether other IMD pathway elements are involved,

OR

(2) by a fuller discussion of the unusual nature of Relish's contribution to restraining hematopoiesis -- meaning that such an activity seems "anti-inflammatory" (tho some would disagree with this interpretation), which is opposite to the typical role that Relish and other NF-kB play. Is Relish's role dependent on its transcriptional activity or nuclear translocation? Is it similar to NF-kB role in promoting stemness in other tissues/organs/systems? In the various functions of Relish in Drosophila biology, is there any correlation between involvement of Imd pathway and type of functioning? In some ways, the lymph gland is a great tissue to understand the inflammatory vs non/anti-inflammatory functions of Relish, because it is an organ that does have such an important inflammatory function.

**Have all data underlying the figures and results presented in the manuscript been provided?**

Reviewer #1: Yes

Reviewer #2: Yes

PLOS authors have the option to publish the peer review history of their article (what does this mean?). If published, this will include your full peer review and any attached files.

Reviewer #1: No

Reviewer #2: No

---

## [Editor Report · Decision Letter 2]

24 Aug 2024

Dear Dr Mandal,

We are pleased to inform you that your manuscript entitled "The NF-κB Factor Relish maintains blood progenitor homeostasis in the developing Drosophila lymph gland" has been editorially accepted for publication in PLOS Genetics. Congratulations!

Yours sincerely,

Pablo Wappner

Section Editor

PLOS Genetics

Gregory Copenhaver

Section Editor

PLOS Genetics

Comments from the reviewers (if applicable):

**Data Deposition**

http://datadryad.org/submit?journalID=pgenetics&manu=PGENETICS-D-24-00160R2

**Press Queries**

---

## [Editor Report · Acceptance letter]

4 Sep 2024

PGENETICS-D-24-00160R2 

The NF-κB Factor Relish maintains blood progenitor homeostasis in the developing Drosophila lymph gland 

Dear Dr Mandal, 

We are pleased to inform you that your manuscript entitled "The NF-κB Factor Relish maintains blood progenitor homeostasis in the developing Drosophila lymph gland" has been formally accepted for publication in PLOS Genetics! Your manuscript is now with our production department and you will be notified of the publication date in due course.

With kind regards,

Anita Estes

PLOS Genetics

On behalf of:
